



# Assimilation of satellite swaths versus daily means of sea ice concentration in a regional coupled ocean-sea ice model

Marina Durán Moro[1], Ann Kristin Sperrevik[1], Thomas Lavergne[1], Laurent Bertino[2], Yvonne Gusdal[1], Silje Christine Iversen[3], and Jozef Rusin[1]

[1]Norwegian Meteorological Institute, Oslo, Norway
[2]Nansen Environmental and Remote Sensing Center, 5007 Bergen, Norway
[3]Department of Physics and Technology, UiT The Arctic University of Norway, Tromsø, Norway

**Correspondence:** M. Durán Moro (marinadm@met.no)

**Abstract.** Operational forecasting systems routinely assimilate daily means of sea ice concentration (SIC) from microwave radiometers in order to improve the accuracy of the forecasts. However, the temporal and spatial averaging of the satellite individual swaths into daily means of SIC entails two main drawbacks: (i) the spatial resolution of the original product is blurred (specially critical on periods with strong sub-daily sea ice movement), and (ii) the sub-daily frequency of passive microwave

5 observations in the Arctic is not used, providing less temporal resolution in the data assimilation (DA) analysis and therefore, in the forecast. Within the SIRANO (Sea Ice Retrievals and data Assimilation in NOrway) project, we investigate how challenge (i) and (ii) can be avoided by assimilating satellite individual swaths (Level-3 Uncollated) instead of daily means (Level-3) of SIC. To do so, we use a regional configuration of the Barents Sea (2.5 km grid) based on the Regional Ocean Modeling System (ROMS) and The Los Alamos Sea Ice Model (CICE) together with the Ensemble Kalman Filter (EnKF) as the DA system. The

10 assimilation of individual swaths significantly improves the EnKF analysis of SIC compared to the assimilation of daily means; the Mean Absolute Difference (MAD) shows a 10% improvement at the end of the assimilation period, and a 7% improvement at the end of the 7-day forecast period. This improvement is caused by better exploitation of the information provided by the SIC swath data, in terms of both spatial and temporal variance, compared to the case when the swaths are combined to form a daily mean before assimilation.

## 1 Introduction

The Arctic ocean is a complex dynamical region with strong air-sea-ice interactions (Garuba et al., 2020; Deng and Dai, 2022). Climate change has a strong impact on this region leading to a significant reduction of sea ice during the summer period (AMAP, 2021; Thomas et al., 2022). The Barents Sea, a marginal sea of the Arctic Ocean, is particularly sensitive to climate change (Barton et al., 2018; Jørgensen et al., 2022) and has been warming faster than any other place in the Arctic (Lind et al.,

20 2018; Isaksen et al., 2022). The Barents Sea is a part of The Northeast Passage (NEP), one of the main ship transportation routes passing through the Arctic Ocean, connecting the Pacific with the Atlantic Ocean with a shorter distance than the traditional routes (Cao et al., 2022; Min et al., 2022). The increase in the number of ships traveling this route during the summer months due to the Arctic sea ice melting at a faster rate will increase the likelihood of incidents in the Barents Sea. Furthermore, this



strong decline of sea ice promotes the increase of other human activities in the region, mainly related to fishing, tourism, and oil and gas exploration. There will therefore be a growing demand of more accurate sea ice and ocean forecasts in the Barents Sea, in order to avoid any type of hazards such as thick ice, ice-berg drift or poor weather conditions, with a focus on a higher spatial and temporal precision of these forecasts.

Although there has been major advances in the development of ocean and sea ice numerical models in the past decades, there is still a significant number of processes not well understood and not properly represented by the models. In particular, sea ice physics are still not adequately described in numerical models, and investigation on new numerical and physical parameterisations is needed (Blockley et al., 2020). In a forecasting context, it can therefore be beneficial to apply robust data assimilation (DA) methods, such as the Ensemble Kalman Filter (EnKF), to ensure that the model predictions produced by a coupled ice-ocean model are constrained by the available observations (Lisæter et al., 2003). The significant uncertainties of many numerical physical processes (Brankart et al., 2015) have motivated the use of ensemble model simulations to provide uncertainties associated to the forecasts. One of today's main operational forecasting systems of the Arctic ocean is TOPAZ4 (Sakov et al., 2012) which is based on the EnKF. An Ensemble Prediction System (EPS) has recently been implemented for the Barents Sea (Barents EPS, Röhrs et al. 2023) at MET Norway. The Barents EPS uses the *Barents-2.5* configuration (Debernard et al., 2021), which is based on the Regional Ocean Modeling System (ROMS, Shchepetkin and McWilliams 2005) and the Los Alamos Sea Ice Model (CICE, Hunke et al. 2017), to provide hourly forecasts of ocean and sea ice variables, and applies the EnKF to assimilate satellite daily means of sea ice concentration (SIC) and sea surface temperature (SST), as well as in situ data of temperature and salinity. Although the system successfully provides short and medium-range forecasts of the ocean and sea ice in the region, there is a need for improvement, in particular, concerning the forecasts of the marginal ice zone.

Satellites have provided a continuous data record of sea ice measurements in the polar regions since 1979 using different passive microwave sensors (Parkinson, 2022). Passive microwave remote sensing measures the microwave radiation emitted by the Earth's surface and atmosphere at various frequencies and polarisations. The distinct emissivity of ice and open water enables the calculation of SIC. A series of SIC algorithms exist (Ivanova et al., 2015; Kern et al., 2019) which exploit different combinations of frequencies and polarisations. The main advantage of passive microwave remote sensing is its overall independence of cloudiness and light conditions, thus providing data during day and night. On the other hand, a main drawback is that the radiation has to be integrated over a large region due to the low energy emission from the Earth's surface. This leads to a coarse spatial resolution in the final SIC product, and consequently, the loss of small details and features in the sea ice, including leads. One of the more recent passive microwave mission is AMSR2 (Advanced Microwave Scanning Radiometer 2) onboard the GCOM-W1 (Global Change Observation Mission – Water) satellite launched in May 2012. Data from this sensor are currently used to produce data records of SIC maps of the Arctic region, such as the ones provided by OSI SAF[1] and the University of Bremen[2]. Recently, within the SIRANO (Sea Ice Retrievals and data Assimilation in NOrway)[3] project,

---

[1]https://osi-saf.eumetsat.int/products/sea-ice-products

[2]https://seaice.uni-bremen.de/sea-ice-concentration/

[3]https://cryo.met.no/en/sirano





Rusin et al. (2023) have implemented a resolution-enhancing technique which aims at producing SIC maps at a higher spatial resolution than the ones provided by the classic OSI SAF algorithm.

 The satellite individual swaths of SIC show significant sub-daily displacements and changes on the sea ice distribution mainly due to strong winds or rapid temperature changes. The averaging of these swaths to produce daily composited maps blurs the spatial resolution of the satellite products before assimilation, meaning that part of the sub-daily dynamics is lost.

Hence, this does not allow to take advantage of the sub-daily frequency of passive microwave observations in the Arctic. The assimilation of satellite individual swaths, i.e. Level-3 Uncollated (L3U), instead of daily means, i.e. Level-3 (L3), could therefore lead to an improvement of short- to medium-range forecasts of the sea ice distribution. As previous studies have only assimilated daily means of SIC (Sakov et al., 2012; Fritzner et al., 2020; Röhrs et al., 2023), we aim in this study to explore the benefits of the assimilation of individual swaths of SIC in order to better constrain the sea ice forecasts both spatially and

temporally.

 The *Barents-2.5* numerical model together with the EnKF-C software (Sakov, 2014) are used in this study. Two DA experiments are performed: (i) the first experiment assimilates daily means of SIC (synchronous assimilation); (ii) the second experiment assimilates individual swaths of SIC (asynchronous assimilation, Sakov et al. 2010). In synchronous assimilation, the EnKF assumes that the time of the observation corresponds to the analysis time. In asynchronous assimilation, the EnKF

takes the time of each individual swath into account in the analysis. The satellite observations assimilated are the L3 and L3U SIRANO SICs based on AMSR2 data (Rusin et al., 2023). Assimilation is performed every second day with a 7-day forecast period at the end of the experiment period. Results are validated using independent ice-charts data (JCOMM, 2017).

 The article is organized in 5 sections. In Sect. 2, the numerical model and its regional setup are presented as well as the DA technique used for the experiments. The DA experiments are described in Sect. 3. Section 4 presents the results obtained from

the DA experiments, followed by a more detailed discussion in Sect. 5. Finally, conclusions and perspectives are provided in Sect. 6.

## 2 Region of study and numerical tools

### 2.1 Region of study

The Barents Sea and waters around Svalbard are the areas of interest in this study, with the exact region being defined by the

geographical limits of the *Barents-2.5* numerical domain. Figure 1 presents the *Barents-2.5* domain along with the numerical bottom topography. The numerical domain covers the Barents Sea, the northern Norwegian Sea, and areas of the Greenland Sea and Arctic Ocean near Svalbard and Franz Jozef Island. The deepest areas (>3 km) are situated in the north-west of the domain outside the Barents Sea, the latter being a rather shallow shelf sea with an average depth of ∼230 m and a maximum depth of ∼500 m at the western end of Bear Island Trough. The Barents Sea topography is characterized by troughs and basins,

separated by shallow bank areas. This region serves as a passage for Atlantic water to reach the Arctic ocean, with warm and saline North-Atlantic water entering through the Barents Sea Opening where it meets colder waters coming from the Arctic Ocean (Loeng, 1991). This encounter causes the formation of the Polar Front which plays a major role in the formation of



sea ice and consequently, in the biological activity in the area (Våge et al., 2014). North-Atlantic waters also reach the Arctic Ocean through the West Spitsbergen Current which traverses the Fram Strait following the west coast of Svalbard. Along
both these different pathways, Atlantic water becomes cooler and fresher. It cools down in the ice-free areas by heat loss to the Arctic atmosphere, and becomes fresher when mixing with the fresh surface layer in the marginal ice zone (Jones, 2001). In the Barents Sea, there is a seasonal ice cover with a maximum extent reached in March/April and a minimum in August/September (Mohamed et al., 2022).

## 2.2  Numerical model system

In this study, we use the *Barents-2.5* configuration based on the METROMS (Kristensen et al., 2017; Debernard et al., 2021) framework[4]. This framework has been successfully applied in several studies in the polar regions (Naughten et al., 2018; Fritzner et al., 2019, 2020; Röhrs et al., 2023). METROMS uses ROMS as the ocean component, and CICE as the sea ice component. The two numerical models are coupled applying the Model Coupling Toolkit technique (MCT, Larson et al. 2005). In the work presented here, ROMS version 3.7 and CICE version 5.1.2 are used. ROMS is a 3-dimensional (3D) ocean model
that solves the hydrostatic primitive equations using the Boussinesq approximation in a three-dimensional grid. ROMS uses a stretched terrain-following $\sigma$-coordinate in the vertical, and an Arakawa-C grid in the horizontal dimension. CICE simulates the growth, melt and movement of sea ice. The momentum equation and thickness advection are solved on a quadrilateral Arakawa B-grid. Four main interacting components constitute the CICE model: a vertical thermodynamic model, an elastic-viscous-plastic model, an incremental remapping transport model, and a ridging parameterization.

The *Barents-2.5* model has been used in the regional SIC prediction study by Fritzner et al. (2020) and in the Barents EPS implemented by Röhrs et al. 2023. The *Barents-2.5* model (see model domain in Fig. 1) employs a curvilinear grid in the horizontal with a 2.5 km resolution (varying slightly throughout the numerical domain), and has 42 vertical layers. The bottom topography is taken from the GEBCO global data set and interpolated to the model grid. We collect TOPAZ4 operational data[5] from the Copernicus Marine Environment Monitoring Service (CMEMS), which is then interpolated onto the *Barents-*
*2.5* grid and used for the boundary conditions for ocean and ice variables. Tidal information is obtained from TPXO7.2 tidal model (Egbert and Erofeeva, 2002). River runoff climatology is collected from the Norwegian Water Resources and Energy Directorate (NVE)[6] data for mainland Norway and the HYPE (Hydrological Predictions for the Environment) (Lindström et al., 2010) model for Svalbard and Russia. Sea ice is modelled in 5 discrete thickness categories, each with 7 vertical layers and a single snow layer on top. More technical details on the *Barents-2.5* model setup are described in Duarte et al. (2022) and Röhrs
et al. (2023).

---

[4]https://github.com/metno/metroms.git
[5]https://data.marine.copernicus.eu/product/ARCTIC_ANALYSIS_FORECAST_PHYS_002_001_a/description
[6]https://www.nve.no/vann-og-vassdrag/hydrologiske-data/




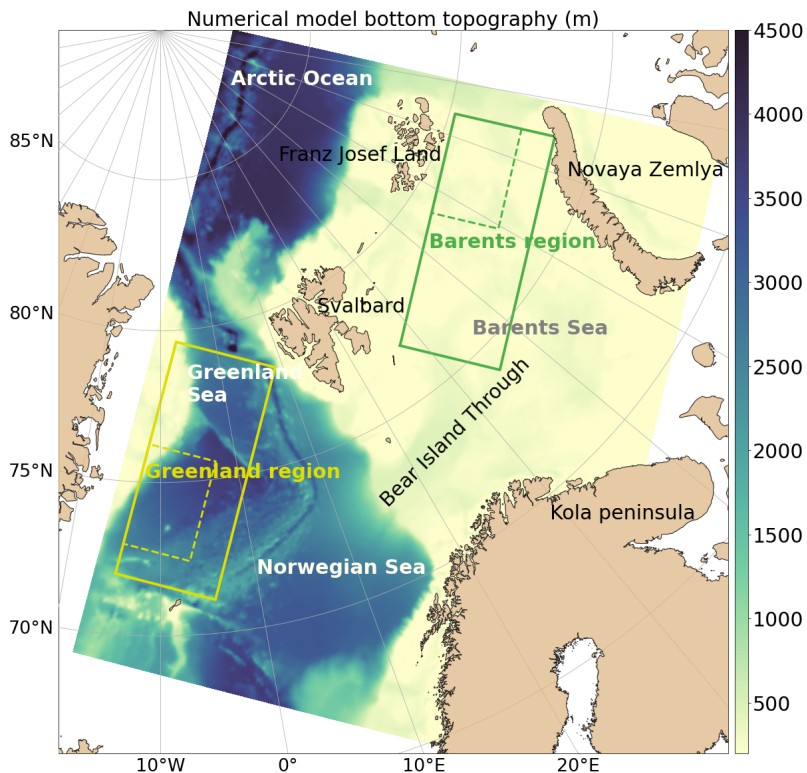

**Figure 1.** *Barents-2.5* numerical model domain and bottom topography (m). The green and yellow rectangles indicate smaller regions selected for a more detailed analysis of the results.

## 2.3 Data Assimilation system

The EnKF (Evensen, 1994; Burgers et al., 1998; Houtekamer and Mitchell, 1998; Evensen, 2003) is selected to perform the DA in this study. The EnKF is based on the Kalman Filter (KF) equations (Kalman, 1960). The KF analysis equations are:

$$x_a = x_b + K(y - Hx_b)$$
$$K = P_b H^T (H P_b H^T + R)^{-1},$$

(1)

with $x_a$ being the new corrected model state, $x_b$ the background model state, $K$ the Kalman gain, $y$ the observations, $H$ the observation operator, $P_b$ the background error covariance matrix, and $R$ the observation error covariance matrix. The new corrected state, $x_a$, is found through a global nonlinear minimisation problem; this corrected state minimizes a cost function that measures the difference between the observations and the background state with a recall background term. The EnKF is based on the KF equations, however, for the case of EnKF, there is an ensemble of $m$ model states $E$. This ensemble is defined

by its ensemble mean $x$ and anomalies $A$. Then, the EnKF can be related to the KF equations [see Eq. (1)] through these





definitions:

$$\boldsymbol{x} = \frac{1}{m}\boldsymbol{E}\mathbf{1}$$
$$\boldsymbol{P} = \frac{1}{m-1}\boldsymbol{A}\boldsymbol{A}^T \tag{2}$$
$$\boldsymbol{A} = \boldsymbol{E} - \boldsymbol{x}\mathbf{1}^T,$$

with $\mathbf{1}$ a vector with all elements equal to 1. The EnKF algorithm computes a vector of linear coefficients for updating the mean and an ensemble transform matrix that updates the ensemble anomalies. Within the EnKF-C software Sakov (2014),

we select the Deterministic Ensemble Kalman Filter (DEnKF, Sakov and Oke 2008) scheme to perform the DA analysis. By assuming a small $\boldsymbol{KH}$ term, the DEnKF uses half of the Kalman gain for updating the ensemble anomalies which provides various advantages against other EnKF schemes. The DEnKF has been successfully used in several coupled ice-ocean models (Cheng et al., 2023).

Concerning the localisation methods, the EnKF-C code uses local analysis which consists of the calculation of local ensem-

ble transforms for each grid cell of the state vector. The system uses only horizontal localisation: the assimilation computes a common horizontal array of local ensemble transforms and then applies them to each horizontal field of the model. The local analysis uses only local observations within a certain localisation radius of the corresponding grid cell. The localisation radius is defined by the numerical entry $r_{loc}$, whose value defines the support localisation radius in $km$. When we increase this radius, the number of local observations increases leading to a stronger impact of the observations in the analysis. However, large

increases of this radius can deteriorate the analysis caused by spurious correlations. To decrease the impact of the observations, the numerical entry $R-$factor can be increased. This parameter acts as a scaling coefficient of the observation error variances in the $R$ matrix. An inflation factor can be defined to increase the ensemble anomalies with a capping rule which restricts inflation in zones where spread is barely modified after assimilation.

The Degrees of Freedom of Signal (DFS) is an observation impact metric provided by the EnKF-C. The DFS can be used as

an indicator of rank problems. The recommendation is that this metric stays within a range between half and a quarter of the total ensemble size. The DFS can be decreased by reducing the $r_{loc}$, which will reduce the number of observations used in the analysis. Another observation metric provided is the Spread Reduction Factor (SRF) which informs on the strength of the DA, in particular on the ensemble spread reduction after analysis. The SRF values should stay below 1 in order to remain close to the assumed linear regime; an increase of the $R-$factor will reduce the SRF due to a higher observation error variance.

The EnKF-C offers two assimilation modes, (i) synchronous and (ii) asynchronous, which differ on the temporal account-ability made of the observations during the analysis. In the synchronous mode, the observations are assumed to be made at the time of assimilation, and the EnkF-C code does not use the time information in the observation file. The asynchronous mode takes into account the specific time of the observations (Sakov et al., 2010) in the analysis. The model estimation for the respective observation is performed by using the background model state at the pertinent time. Hence, in asynchronous mode,

both the time information in the model state and observations are used.





## 2.4 Validation and other metrics

Different computations are presented below which are used to validate the model output with the observations.

The Root Mean Square Error (RMSE) between model data and observations is computed following this equation:

$$\text{RMSE}(X,Y) = \sqrt{\frac{1}{m}\sum_{k=1}^{m}(X_k - Y)^2}, \tag{3}$$

with $X$ the model state, $Y$ the observation, $k$ the ensemble member number and $m$ the ensemble size. This RMSE computation provides a RMSE map which can be spatially averaged to obtain a total mean RMSE value.

The standard deviation (std) defined below provides a spread map of the ensemble:

$$\text{std} = \sqrt{\frac{1}{m}\sum_{k=1}^{m}(X_k - \overline{X})^2}. \tag{4}$$

A spatial average from both previous metrics can be computed following $\overline{A} = 1/N \sum_{i=1}^{N} A_i$ with $N$ the total number of spatial
points.

In order to compare model data with observations, we define a Mean Absolute Deviation (MAD):

$$\text{MAD}(X,Y) = \frac{1}{N}\sum_{i=1}^{N}\left(\frac{1}{m}\sum_{k=1}^{m}|(X_k(i) - Y(i))|\right), \tag{5}$$

which averages the absolute difference between model data and observations at a fixed time step.

The Data Assimilation Skill Score (DASS) measures the improvement provided by the analysis compared to the background
state. Following the work in Ren et al. (2016), we define here the DASS as:

$$\text{DASS} = 1 - \frac{\overline{\text{RMSE}(X_a,Y)}}{\overline{\text{RMSE}(X_b,Y)}}, \tag{6}$$

with $\overline{\text{RMSE}(X,Y)} = 1/N \sum_{i=1}^{N} \text{RMSE}(X_i,Y_i)$. $X_a$ and $X_b$ are the analysis and background states, respectively. When DASS is positive, it means that data assimilation is improving the forecast state. When the score is negative, the data assimilation is contaminating the forecast state: we deteriorate the background state instead of improving it.

## 2.5 Power spectra

Spectral analysis of ocean or sea ice variables provides information on the level of variance present at different spatial scales. The computation of spectral analysis of modelled SIC before and after assimilation can therefore provide information on how the SIC field is modified in terms of its spatial variability. A general reduction of the spectral density indicates that the SIC field has become more homogeneous with less spatial power. If the density reduction is constrained at specific wavelengths, it
means that at those wavelengths there is a decrease of the quantity of spatial features. Areas fully ice-covered (SIC$= 100\%$) or open water areas (SIC$= 0\%$) are characterized by zero spectral density of SIC as there is no spatial variability.



Modelled and observed data are usually decomposed into spectral space using the discrete Fourier transform (DFT). Different methods can be applied over the data to make them periodic which is a requirement for the DFT computation. However, these methods can modify the final spectrum. In order to avoid this, an alternative spectral analysis method based on the discrete cosine transform (DCT) is chosen for this study. The reader is referred to Denis et al. (2002) for a thorough description of the DCT. This technique has recently been applied for different variables, such as the SST in Iversen et al. (2023), SIC in Rusin et al. (2023) or surface velocities in Arango et al. (2023). The DCT technique is applied together with the updated weighted wavelength binning by Ricard et al. (2013).

## 3   Description of DA experiments

### 3.1   General setup of the assimilation experiments

Sea ice and ocean dynamics in the Barents Sea region are simulated using the *Barents-2.5* model presented in Sect. 2.2. The forecasts provided by the model are updated through the assimilation of SIRANO SIC observations. First, the ensemble members are run for 3 months (January-March 2022) to reach a stable dynamical state. The initial ocean and sea ice conditions are derived from the same TOPAZ4 operational data used to extract the boundary conditions (see Sect.2.2). Then, we perform three different experiments for the month of April 2022: (i) CTRL; (ii) SYN; and (iii) ASYN. The CTRL experiment is a free run; the SYN experiment synchronously assimilates daily means of SIC (L3); and the ASYN experiment asynchronously assimilates individual swaths of SIC (L3U). The assimilation period is from the 3 to the 23 April 2022 with assimilation being carried out every second day. There is therefore a total of 11 EnKF analysis with an inter-analysis period in between them when no DA is performed. This inter-analysis period enables a stronger increase of the model spread before the next analysis. After the 23 April 2022, no more DA is carried out and a 7-day forecast is provided.

Ensemble spread is created through the use of different atmospheric forcings. In this study, two sources of atmospheric data are selected to force the *Barents-2.5*: (i) the Integrated Forecast System (IFS, Owens and Hewson 2018) developed at the European Centre for Medium Ranged Weather Forecast (ECMWF); and (ii) the Numerical Weather Prediction (NWP) system AROME-Arctic (MET-AA) developed at MET Norway (Müller et al., 2017a, b). ECMWF provides an atmospheric forecast ensemble of 51 members (ECMWF-ENS[7]) with a resolution of 18 km. MET-AA provides a deterministic forecast at high-resolution (2.5 km) and it has the great advantage of being on the same domain and resolution as the *Barents-2.5*. The atmospheric forcing used in the experiments consists of hourly surface fields of wind, air temperature, pressure, relative and specific humidity, rain fall rate and cloud fraction. A model ensemble of 6 members is run through the entire period, including the 3 months spin-up period. Members 1 to 5 are forced by 5 different members from ECMWF-ENS, and member 6 is forced by MET-AA. The ensemble used is therefore not identically distributed because the MET-AA member can be quite different from the other 5 members. An ensemble size of 6 members allows us to investigate the differences between the two DA experiments while keeping computational and data storage costs reasonable. Although a small ensemble may be at risk of instabilities for

---

[7]https://www.ecmwf.int/en/forecasts/datasets/set-iii





long integrations due to spurious correlations, in our study an ensemble size of 6 members is a fair choice as we only perform
11 EnKF analysis during the assimilation period and only surface correlations between the SIC fields of each ice category and
the aggregated value are used.

Daily model outputs at 00 UTC for each ensemble member are updated through the EnKF-C software using the SIRANO
SIC observations available during the previous 24 hours. The model provides outputs of SIC for 5 ice categories, while the
aggregate of these are corrected during the assimilation. The 5 ice categories are then updated through the correlations with
the aggregated value given by the background ensemble covariance matrix. After the assimilation, there is a post-processing
step where other ice variables are set to zero where there is open water. The final updated model field is used to initialize the
following inter-analysis run. In the experiments presented here, ocean fields are not modified by the assimilation, and their
background and analysis fields are therefore identical. EnKF-C numerical parameters are equally defined within the SYN and
ASYN experiments: $r_{loc} = 20$ km; $R$-factor $= 20$; and $K$-factor $= 2$. An inflation factor of $10\%$ is used. EnKF-C diagnostics
(DFS and SRF) are checked after every assimilation analysis to verify that they are within the defined ranges (see definitions
in Sect. 2.3).

In both the SYN and ASYN experiments the model SIC field is updated at 00 UTC. In the SYN experiment, the assimilated
observations are the SIRANO daily means of SIC (L3). In the ASYN experiment, each 24 hours assimilation window is split
into 4 sub-windows, each with a range of 6 hours. Each individual satellite swath (∼11 swaths per day) contains information
on the observation time and ensures that each swath is used in the appropriate sub-window. A total of 5 model states con-
taining the model SIC fields are provided to the EnKF: one corresponding to the model output at 00 UTC and the others are
the 6-hourly averages computed during the previous 24 hours. The last four model files contain the model time centered at
each corresponding observation window: 21 UTC, 15 UTC, 09 UTC and 03 UTC. Then, the final model output at 00 UTC is
corrected using the correlation coefficients from the updated covariance matrix.

## 3.2   Background model ensemble

Background model surface ensemble means of SIC and thickness (SIT) on the 3 April 2022 at 00 UTC are shown in Fig. 2a and
Fig. 2b respectively. The two colored rectangles in these figures indicate two smaller regions where we perform a more detailed
investigation of the results (also indicated in Fig. 1): (i) the Barents region located in the northern part of the Barents Sea (green
rectangle); (ii) the Greenland region located in the Greenland sea (yellow rectangle). Both regions have the same shape and
size ($150 \times 350$ grid cells) and the ice edge is contained inside both of them during the entire period of the study. The ice edge
area is subject to rapid daily and sub-daily ice changes, and thus is where accurate DA systems are mostly needed. The SIC
mean value is similar within both regions, however the Barents region presents much thinner ice ($\overline{\mathrm{SIT}} = 0.56$ m) compared to
the Greenland region ($\overline{\mathrm{SIT}} = 1.61$ m). The latter region receives multi-year ice outflow through Fram Strait, while the Barents
region presents almost exclusively thinner first-year ice. This thinner ice is more dependent on local atmospheric conditions
resulting in faster freezing/melting dynamics. Figure 2c corresponds to the difference between the mean background ensemble
of SIC (see Fig. 2a) and the SIRANO daily mean observations (see Fig. 3a). The background state has too much ice compared
to the observations, with the strongest errors located around the ice edge area (dark red zones). Some pronounced differences


are also seen between Svalbard and Franz Josef Land due to areas with low SIC in the observations (see Fig. 3). Background differences are higher in the Greenland region ($\overline{\text{diff}_{\text{SIC}}} = 0.48$) than in the Barents region ($\overline{\text{diff}_{\text{SIC}}} = 0.28$). Background model spread of SIC is shown in Fig. 2c. Large values of spread are located within the ice edge area, where the model has more uncertainty. The Greenland region exhibits about twice as large spread as the Barents region. This indicates that the model has stronger uncertainties in the Greenland region and consequently, ensemble members bear less resemblance to one another.

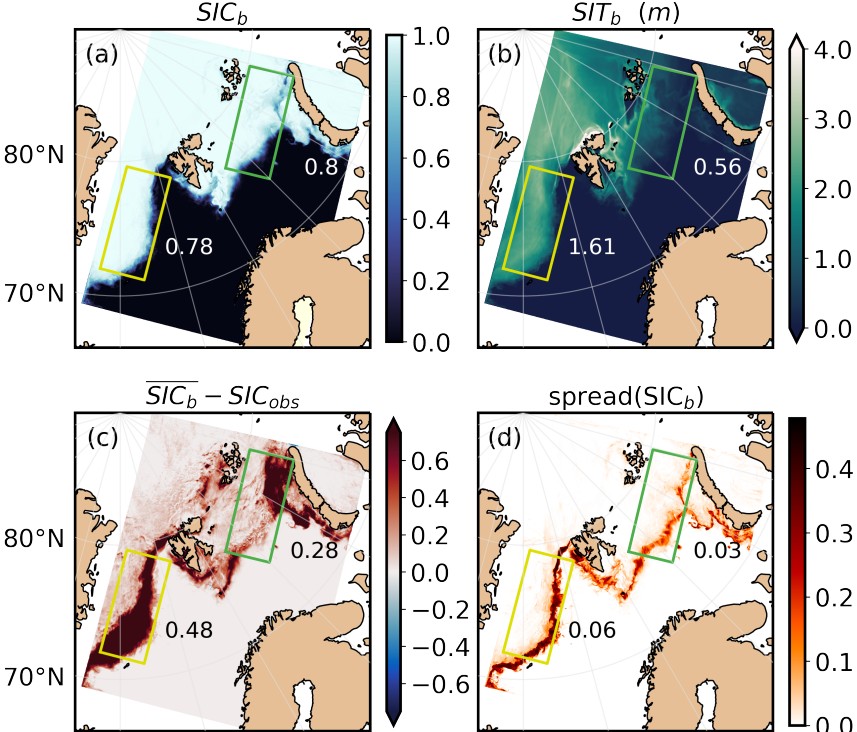

**Figure 2.** Means of (a) SIC and (b) SIT (m) from the background ensemble; (c) mean difference between the background ensemble and the satellite daily mean (L3) of SIC; and (d) background model spread of SIC. Model data corresponds to output on the 3 April 2022 at 00 UTC. L3 observation corresponds to the daily mean on the 2 April 2022. Barents and Greenland regions are indicated by the green and yellow rectangles respectively.

### 3.3   SIC observations used for assimilation

SIC observations from the AMSR2 mission are the only data source assimilated in the experiments presented in this study. AMSR2 is a dual-polarized, conically scanning, microwave radiometer with a swath width of ~1450 km. This instrument records Earth leaving microwave radiation at seven frequencies from 6.9 to 89.0 GHz, with the latter having the highest spatial resolution (~5 km). The SIC observations and uncertainties assimilated in our experiments are produced using the algorithm implemented by Rusin et al. (2023) in the context of the SIRANO project. The SIRANO SIC fields are the result of applying a pan-sharpening algorithm on two SIC fields: a low-resolution (~15 km) SIC field with low uncertainty obtained from the 18.7



and 36.5 GHz imagery channels, and a high resolution (∼5 km) SIC field with higher uncertainty produced using the 89 GHz

imagery channels. The combination of these two fields through the pan-sharpening method results in SIC fields that have ∼5 km resolution and low uncertainties. The total SIC uncertainty consists of two components: (i) the inherent uncertainty of the SIC algorithm, and (ii) the representativeness uncertainty. More details on the uncertainty computation can be found in Lavergne et al. (2019). The reader is referred to the discussions in Rusin et al. (2023) for details on the algorithm and its evaluation. In our study, this SIC algorithm is used to prepare SIRANO SIC fields and uncertainties both as individual orbits (L3U), and

as daily average maps (L3) directly on the 2.5 km spacing grid of the forecast model. This simplifies the assimilation setup as the observation operator $H$ is then equal to the identity. L3 SIC and uncertainty correspond to the temporal average fields computed from the L3U swaths.

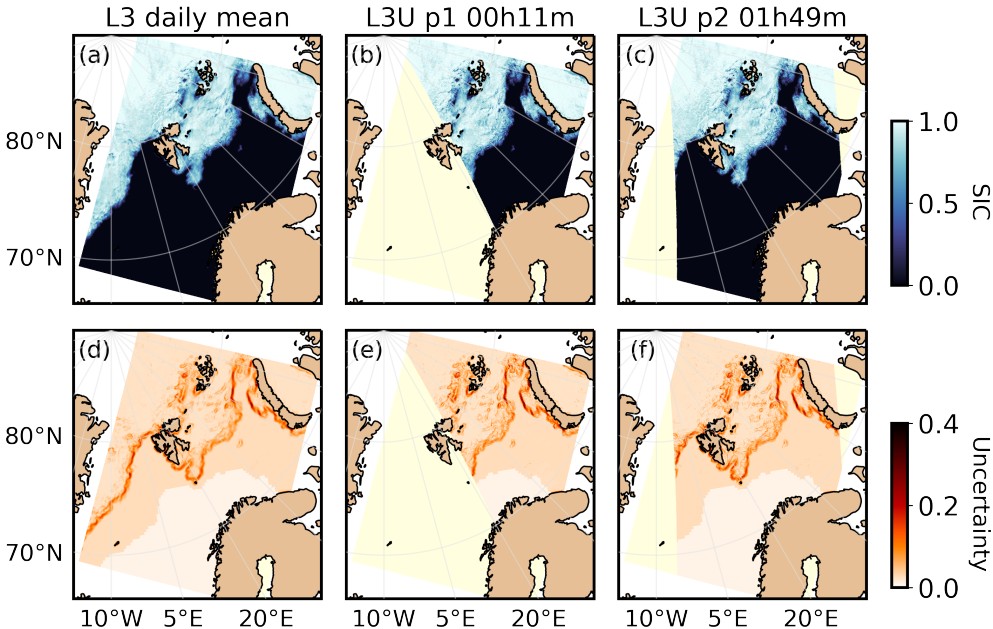

**Figure 3.** SIRANO SIC (a) daily mean (L3), (b) first and (c) second satellite individual swath (L3U), and (d-f) respective uncertainties on the 2 April 2022. A total of 11 swaths are available in the region (see Fig. A1).

Figure 3 shows maps of the SIRANO SIC observations along with their corresponding uncertainty on the 2 April 2022. We present the SIRANO daily mean as well as the two first individual swaths among a total of 11 per day in the *Barents-2.5*

domain (see all swaths in Fig. A1). We observe that the highest uncertainties are in the ice edge area and around water openings in the ice covered areas, with values around 0.2. When comparing the SIRANO daily mean with the corresponding background average field of SIC (Fig. 2a), we observe that: (i) the open water located in the Barents region in the observations is absent in the model field; (ii) a large part of the modelled SIC in the Greenland region is not present in the observations. As presented in Fig. 2c, these two regions are areas with strong differences between model and observations, and where model spread and

observed uncertainty are significantly higher compared to open water or fully ice-covered areas. Our aim is thus to remove ice



from the model state through the assimilation, particularly around the ice edge, in order to achieve better agreement with the observations.

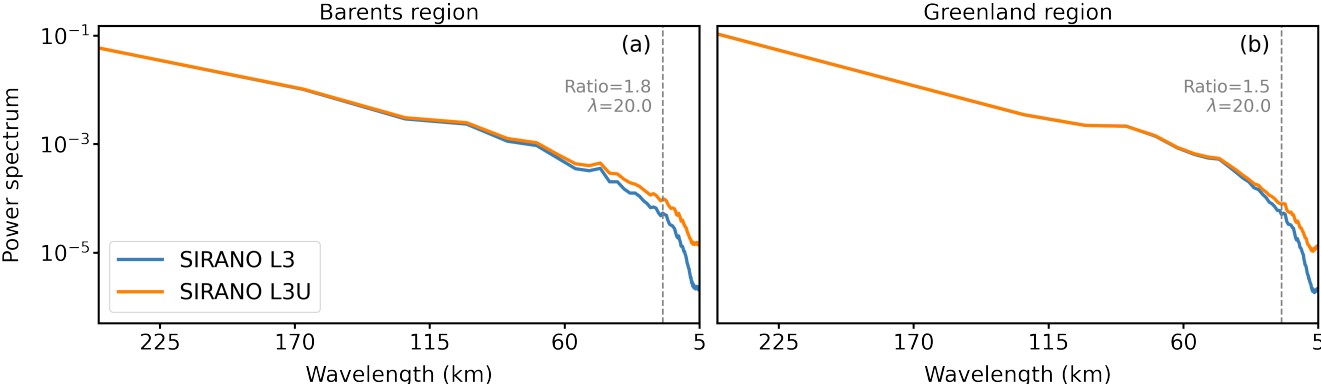

**Figure 4.** Power spectra computed from L3 (blue) and L3U (orange) SIRANO observations for the (a) Barents and (b) Greenland subregions (indicated by dashed rectangles in Fig. 1). Only L3 and L3U observations used in the assimilation experiments (EnKF analysis every second day) are included in the spectra computation. The ratio of L3U to L3 spectrum is indicated for 20 km wavelength.

The power spectrum provides information on the amount of details and small scale features present in the SIC fields. To focus on ice covered areas, two subregions within the Barents and Greenland regions, which are are ice covered during the DA
experiments, are defined (see dashed rectangles in Fig. 1). As we want to investigate how the assimilation of L3U and L3 SIC observations affects the model power spectra in ice-covered regions, it is important to avoid open water regions, which have zero SIC variance and may hinder the interpretation of the results.

Power spectrum from SIRANO data are computed within these two subregions to compare the SIC variance in the L3 and L3U data. The power spectra computed from all SIRANO individual swaths (L3U) and daily means (L3) assimilated in the
experiments are shown in Fig. 4. In both subregions, the L3U power spectra presents in average higher values than the L3 spectra, with the most pronounced difference observed for smaller wavelengths. For wavelengths above 50 km, the ratio to L3 is ∼1; and for wavelengths below 50 km, this ratio increases reaching values close to 7 for wavelengths of 5 km. In particular, the Barents region shows ratios above 1.2 below ∼55 km wavelength, whereas this happens below ∼30 km for the Greenland region. Because the SIRANO L3 daily means are temporal averages of the SIRANO L3U swaths, the L3 maps are smoother
and do not capture the details that are present in the L3U swaths. The spectra calculated for the L3 maps thus contain less spatial power, in particular for small wavelengths.

## 3.4 SIC observations used for validation

Ice-charts from the Ice Service of the Norwegian Meteorological Institute[8](JCOMM, 2017) are used to validate the assimilation experiments. These charts cover the Atlantic sector of the Arctic, focusing on Svalbard and the Barents Sea, and thus provide

---

[8]https://cryo.met.no/en/latest-ice-charts





a full coverage of the *Barents-2.5* domain. The charts are produced on a daily basis, with the exception of weekends and public holidays. They are based on morning passes from multiple satellite sources, primarily synthetic-aperture radar (SAR) data, and have a spatial resolution of less than 1 km. They are considered mostly independent from the assimilated SIC observations because they are derived manually from different satellite instruments by trained analysts. They consist of polygons delineating areas of similar sea ice cover with the following classes: (i) fast ice (FI, SIC = 1.0); (ii) very close drift ice (VCDI,

$0.9 \leq SIC < 1.0$); (iii) close drift ice (CDI, $0.7 \leq SIC < 0.9$); (iv) open drift ice (ODI, $0.4 \leq SIC < 0.7$); (v) very open drift ice (VODI, $0.1 \leq SIC < 0.4$); and (vi) open water (OW, $0.0 \leq SIC < 0.1$). Then, the original polygons are remapped on a 1 km polar stereographic grid. Each SIC interval is indicated by a fixed SIC value in the gridded product corresponding to the middle value of the interval. The ice-chart data are collected through CMEMS[9]. For validation purposes, the ice-charts are interpolated to the model resolution of 2.5 km using a bicubic method. As CICE does not differentiate between stationary ice attached to land

and ice that is floating freely, the FI of the ice-charts maps is considered as VCDI in the validation. Maps of ice-classes are prepared from SIRANO and model SIC data, using the same sea ice classification as the one applied to the ice-chart data.

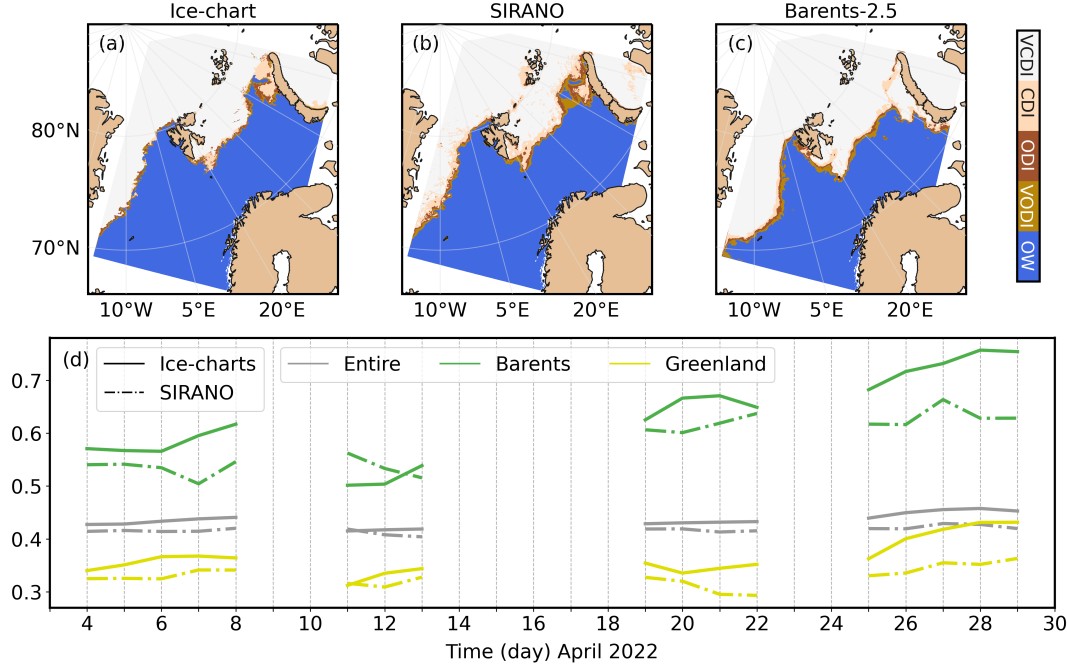

**Figure 5.** Ice-classes maps from (a) ice-charts, (b) SIRANO L3 and (c) CTRL data on the 4 April 2022, and (d) time series of spatial averages of SIC computed daily during 3-30 April 2022. These series are computed for ice-charts (solid lines) and SIRANO ice-classes maps (dash-dotted lines) within the entire domain (gray), the Barents (green) and Greenland (yellow) regions.

To illustrate the resulting ice-classes maps, Fig. 5 shows the ice-chart, the SIRANO and *Barents-2.5* (CTRL data) ice-classes maps computed for the 4 April 2022. The ice-chart and SIRANO maps are similar with slightly more SIC in the ice-chart and

---

[9]https://data.marine.copernicus.eu/product/SEAICE_ARC_SEAICE_L4_NRT_OBSERVATIONS_011_002/description





with a higher presence of the CDI class in the SIRANO map. In contrast, the model ice-classes map differs strongly from the
observed maps by having too much ice coverage, with the highest differences located around the ice-edge area. In the Barents
and Greenland regions, the model map presents a mean SIC above 0.75, whereas the ice-chart and SIRANO map present a
mean SIC of around 0.60 and 0.35 in each region respectively; these comparisons indicate again the need of ice removal from
our background state. Figure 5 also presents a time series of the SIC daily average for 3-30 April 2022 computed from the
ice-charts and SIRANO ice maps in the entire model domain and in both regions. Missing dates correspond to days where
ice-charts are not available. Ice-charts present larger ($\sim 4\%$) mean concentrations than SIRANO. Highest differences ($\sim 7\%$)
are found during the forecast period (25-29 April 2022).

## 4 Presentation of results and validation

### 4.1 Evolution of model variables

Figure 6 presents time series of daily averages of SIC and spread in the Barents and Greenland region for the CTRL, SYN
and ASYN experiments. Both DA series show lower SIC than the CTRL series due to the SIC reduction induced by each
EnKF analysis. During the inter-analysis period, the DA series present a similar behaviour as the CTRL series. In the Barents
region we observe a SIC reduction of 4.4% for the SYN experiment and of 5.5% for the ASYN experiment at the end of
the assimilation period compared to the CTRL. At the end of the forecast period, the SIC reduction falls to 2.2% for both
experiments. In the Greenland region, the SIC reduction is 19.7% and 25.6% at the end of the assimilation period, and at the
end of the forecast period a reduction of 11.7% and 16.2% is seen in the SYN and ASYN experiments, respectively. Most of
the improvement gained with the ASYN assimilation is thus maintained after a 7-day forecast period in the Greenland region.
The Greenland region has higher initial spread (see Fig. 2d) as well as higher spread throughout the period of study (see
Fig. 6c), with a stronger increase rate in the inter-analysis and forecast periods. The initial model spread on the 3 April 2022 is
the highest of the assimilation period for each region. At each EnKF analysis the spread is reduced with a subsequent increase
during the inter-analysis period. The SYN and ASYN spread are similar both in the Barents and Greenland regions with, in
general, a slightly higher ASYN spread during the first half of the assimilation period. Throughout the 7-day forecast, there
is a continuous increase of the model spread in both regions. During the 11 assimilation cycles [1 cycle = 1 EnKF analysis +
1 inter-analysis (or 7-day forecast) period] the ensemble spread does not collapse, confirming that the $R$-factor and inflation
settings are well tuned in the experiments.

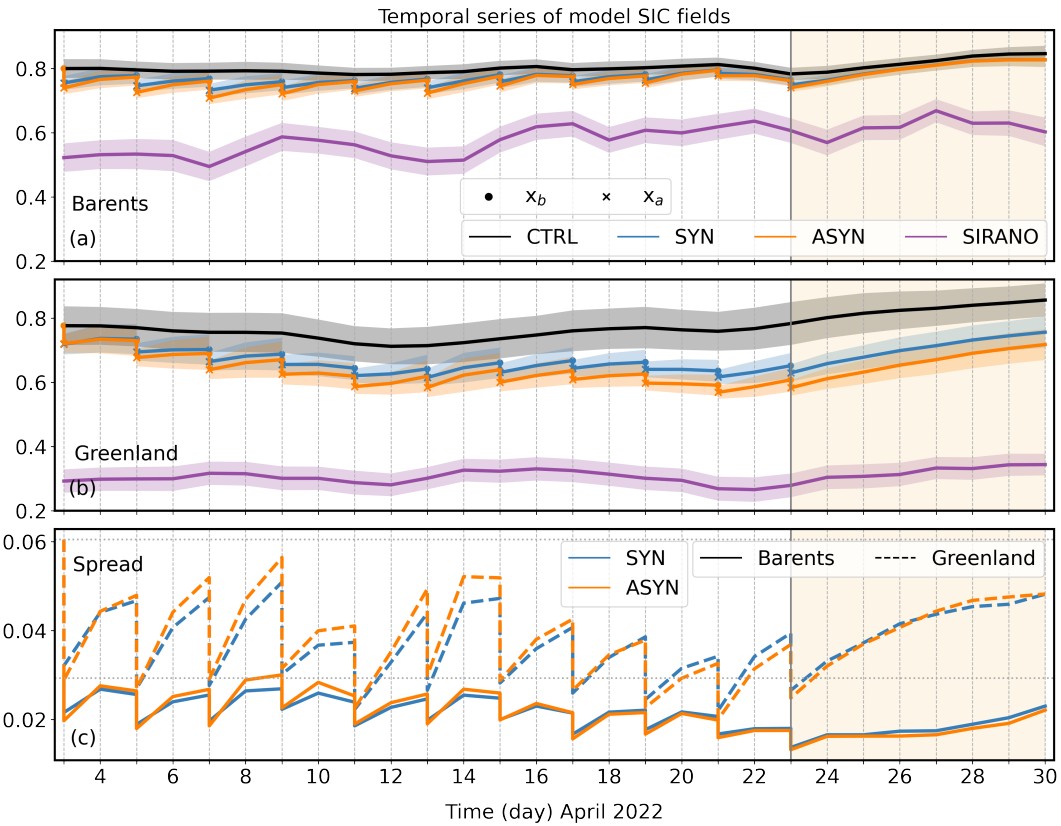

**Figure 6.** Time series of modelled SIC ensemble means from the CTRL (black), SYN (blue) and ASYN (orange) experiments as well as SIRANO L3 data (purple) computed for the (a) Barents and (b) Greenland regions, with the shadowed colored areas corresponding to the ensemble spread and observations uncertainty; and (c) time series of average spread from the SYN and ASYN ensembles within the Barents (solid) and Greenland (dashed) regions. The vertical gray line indicates the start of the forecast period highlighted by a beige shadow.

The SIRANO SIC series presented in Fig. 6a and Fig. 6b indicate that SIRANO data contains lower SIC values than the model, with the strongest differences in the Greenland region. Although the DA experiments present a stronger decrease of SIC in the Greenland region compared to the Barents region, the model errors stay higher in the former due to higher initial background errors (see also Fig. 2c). SIRANO series show a stronger temporal variability in the Barents ($std = 0.045$) compared to the Greenland ($std = 0.020$) region. In contrast, CTRL series show a higher temporal variability in the Greenland region ($std = 0.039$) than in the Barents ($std = 0.017$) region. To illustrate this difference between SIRANO and model temporal variability, Fig. 7 presents SIRANO and CTRL SIC maps at 3 different dates in April 2022. The SIRANO observations show a Barents region with rapid strong freezing/melting processes occurring in a daily basis, whereas the Greenland region has more steady ice dynamics with a similar ice edge location at each date. The CTRL SIC maps show a strong ice coverage with low temporal variability in both regions, and with no presence of water openings in the Barents region. This indicates that



the model over-generates ice in this period of the year and that it does not represent the fast changes occurring in the Barents region.

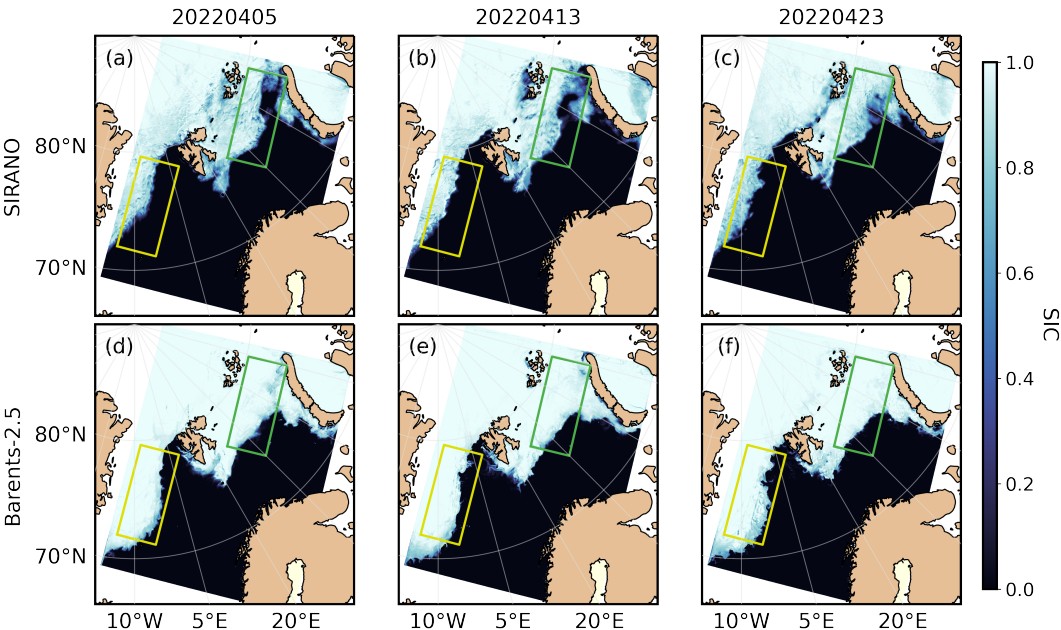

**Figure 7.** Maps of SIC from (a-c) SIRANO L3 and (d-f) model (member 1, CTRL) data on the 5, 13, and 23 April 2022. Barents and Greenland regions are indicated by the green and yellow rectangles respectively.

Solely SIC is assimilated and updated by the EnKF analysis. Other sea ice variables, such as SIT, are updated in a post-processing step in order to ensure consistent sea ice initial conditions for the following inter-analysis run. Figure 8 shows time series of SIT together with series of SST and salinity for the Barents and Greenland regions. DA series show a SIT reduction

and an increase of SST and salinity compared to the CTRL series. In the post-processing step after assimilation, the SIT is set to zero wherever SIC is zero. Because of the removal of ice in many grid points around the ice edge by the EnKF, there is also a decrease of SIT in these areas. Ocean variables, such as SST and salinity, are not modified by the post-processing and the changes in these variables are thus caused by the model response to reduced sea ice coverage.

We observe a higher SIT reduction in the ASYN experiment compared to the SYN experiment in both regions, with a

stronger reduction in the Greenland region due to a stronger sea ice reduction after analysis. In the Barents region, ASYN shows a 4.9% higher reduction of SIT than SYN after the assimilation period, and a 3.8% higher reduction after the forecast period. In the Greenland region, ASYN shows a 13.8% higher reduction of SIT than SYN after the assimilation period, and a 10% higher reduction after the forecast period. There is a larger difference between the two DA experiments in terms of SIT than SIC in both regions. For instance, significant SIT differences (3.8%) are present in the forecast period in the Barents

region, which is not the case for SIC (0%). As the ASYN analysis provides a better correction, a larger open water area is created compared to the SYN analysis. During the inter-analysis period, the model produces new ice over this open water



increasing the SIC. However, this new ice is characterized by low thickness. This mechanism contributes to a lower SIT in ASYN than in SYN during both the inter-analysis and forecast periods in the two regions.

The increase of SST and salinity in the DA experiments is due to the removal of ice in the area, and the consequent increase of
open water in both the Barents and Greenland regions. This new open water progressively becomes warmer and saltier as it is exposed to adjacent sea water and to the atmosphere, which all enhances e.g. mixing processes. In the Barents region, both DA experiments show a $\sim 0.03^{o}C$ SST increase at the end of the forecast period, with a slight larger increase for ASYN. Larger SST increases occur in the Greenland region, with $0.18^{o}C$ increase for SYN and of $0.22^{o}C$ for ASYN at the end of the forecast period. Regarding salinity, the percentage increases of the DA experiments compared to the CTRL are quite low ($< 1\%$) in
both regions; still the ASYN experiment shows higher SSS increases than the SYN one, in particular in the Greenland region. The impact on the SST and salinity fields in the Greenland region is stronger as more ice is removed in this region (see Fig. 6), creating a larger open water area exposed to mixing processes.

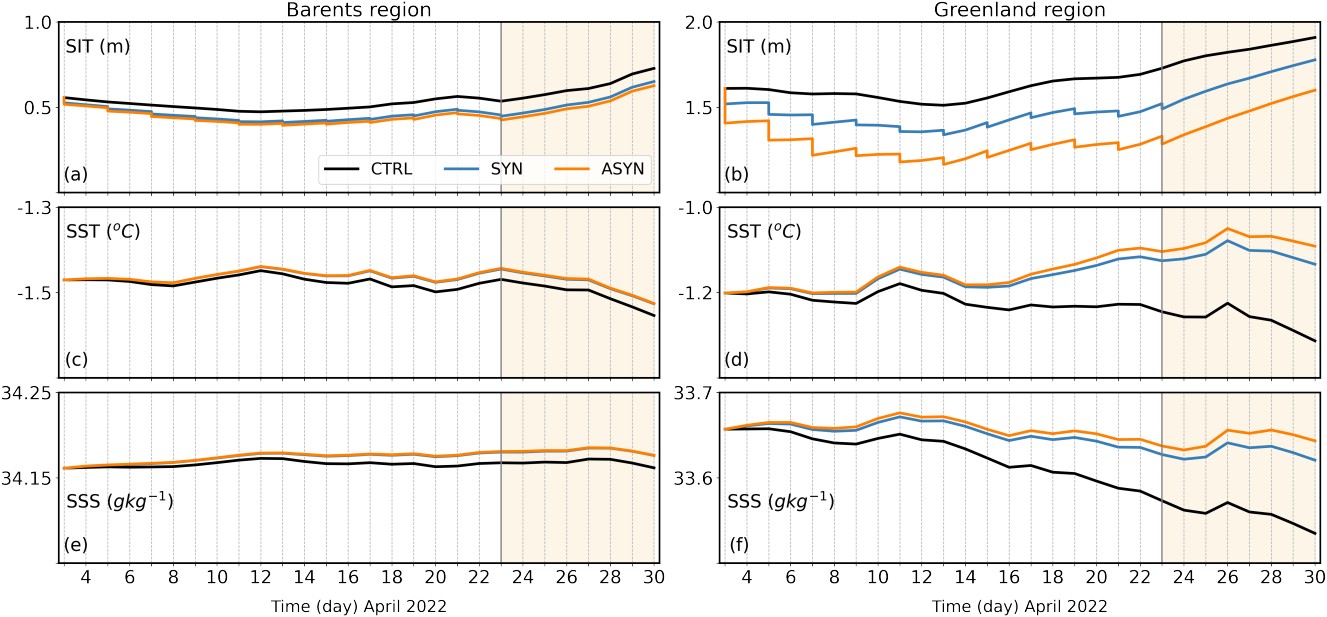

**Figure 8.** Time series of modelled (a, b) SIT, (c, d) SST and (e, f) salinity ensemble means from the CTRL (black), SYN (blue) and ASYN (orange) experiments computed for the Barents and Greenland regions. The vertical gray line indicates the start of the forecast period highlighted by a light orange shadow.

## 4.2   Validation statistics and diagnostics

Figure 9 presents the time series of the MAD [see Eq. (5)] between the model ensemble of SIC and L3 SIRANO data for the
full model domain, as well as the Barents and Greenland regions. This comparison can potentially be more beneficial for the SYN experiment than for the ASYN experiment, as the L3 SIRANO data are assimilated in SYN. The DA experiments present lower MAD values than the CTRL throughout the entire period for all three regions. The MAD decreases after each EnKF

analysis for both DA experiments, as the model state is in better agreement with the SIRANO observations after assimilation. At each EnKF analysis, stronger corrections are provided by the ASYN than by the SYN analysis. During the inter-analysis

periods, the MAD of both SYN and ASYN evolve in a similar way as in the CTRL series.

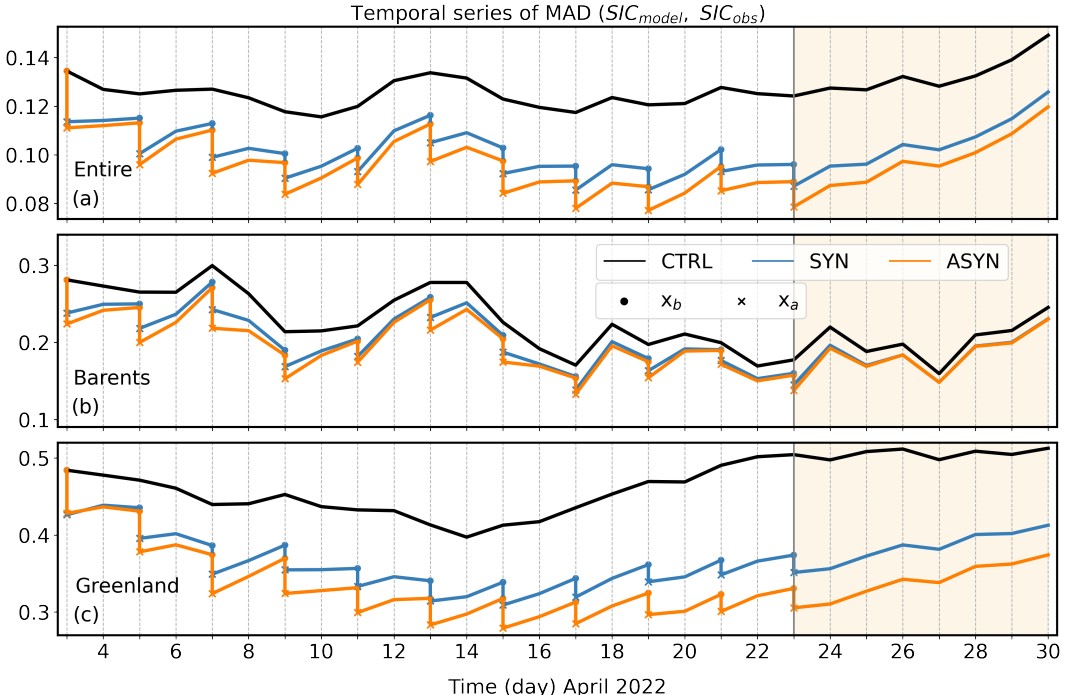

**Figure 9.** Time series of the Mean Absolute Difference (MAD) [see Eq. (5)] computed from SIRANO L3 observations and model data from CTRL (black), SYN (blue) and ASYN (orange) for (a) the entire model domain, (b) the Barents region, and (c) the Greenland region. For SYN and ASYN, the MAD computed from the background and analysis ensemble are indicated by circles and crosses respectively. Between each EnKF analysis there is an inter-analysis period with no DA.

In the MAD series of the full model domain (see Fig. 9a) the difference between the SYN and ASYN experiments is generally maintained during the inter-analysis period, indicating that the improvement provided by the ASYN analysis is maintained after 48 hours. After the last analysis on the 23 April 2022, there is an improvement of 29.9% by SYN and of 36.8% by ASYN compared to CTRL. During the 7-day forecast period, the MAD increases for both DA experiments but still stays below the

CTRL series. On the last day of the forecast (30 April 2022) ASYN provides a 19.8% MAD reduction compared to CTRL, whereas SYN provides a 15.7% reduction. Thus, there is a general better performance of the ASYN experiment compared to the SYN experiment.

Both Barents (Fig. 9b) and Greenland (Fig. 9c) regions show a good EnKF performance with a consistent reduction of the MAD at each EnKF analysis. In the Barents region, the MAD for SYN and ASYN converge by the end of the inter-analysis

period, indicating that the improvement gained by the ASYN analysis is lost after 48 hours. At the end of the assimilation



period, ASYN provides an improvement of 5.1% compared to SYN which is completely lost 48 hours into the forecast period. In the Greenland region on the other hand, ASYN presents considerably lower MADs compared to SYN after the inter-analysis periods. An improvement of 13% is found in ASYN versus SYN at the end of the assimilation period, and the ASYN forecast retains lower MAD than the SYN forecast throughout the 7-day forecast period (11.5% improvement on the last forecast day).

Thus, there is a notably better performance of the ASYN experiment in the Greenland region compared to the Barents region.

The MAD of CTRL experiments evolves differently with time in the Barents and Greenland regions. This temporal variability depends both on fluctuations in modelled and observed data. As noted in Sect.3.2, the model has strong background errors with a general overestimation of ice concentration. High MAD values can therefore be due to increased SIC in the model or caused by decreases in observed SIC not captured by the model. In regions where observations have a high tempo-

ral variability, such as in the Barents region, the fluctuations of the MAD series are governed by the observation variability $[\text{corr}(\text{MAD}_{\text{CTRL}}, \text{SIC}_{\text{SIRANO}}) = -0.93, p < 0.01]$. In regions where the observed SIC exhibit low temporal variability, such as the Greenland region, MAD is governed by the model variability $[\text{corr}(\text{MAD}_{\text{CTRL}}, \text{SIC}_{\text{CTRL}}) = 0.85, p < 0.01]$. MAD series for both DA experiments display similar temporal variability as the CTRL series, indicating that SYN and ASYN results have a strong regional dependence due to both model dynamics and observed variability in the respective regions.

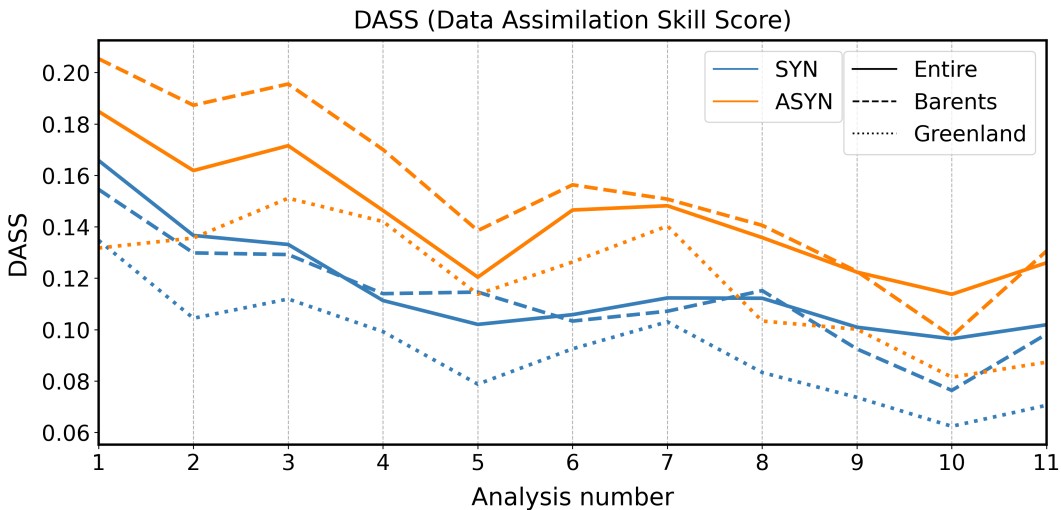

**Figure 10.** Data Assimilation Skill Score [DASS, Eq. (6)] computed for each EnKF analysis for the SYN (blue) and ASYN (orange) experiments. The DASS is computed for the entire domain (solid), for the Barents region (dashed), and for the Greenland region (dotted).

The DASS [see Eq. (6)] is computed for each of the 11 EnKF analyses in order to evaluate the performance of the assimilation. SIRANO L3 observations are used in the DASS computation for both SYN and ASYN. Figure 10 presents the DASS computed for the entire model domain, and for the Barents and Greenland regions. All computed DASSs are positive indicating that the analysis is in better agreement with the observations than the background state. ASYN DASSs are higher than SYN DASSs in all regions (in average $\sim 30\%$ higher), indicating that ASYN performs better than SYN. We also observe that the

DASS decreases during the assimilation period for all experiments and regions; with a decrease of $\sim 37\%$ at the end of the



assimilation period compared with the initial DASS. The DASSs obtained for the Barents region are slightly higher than for the Greenland region, which is related to lower initial background errors in the Barents region.

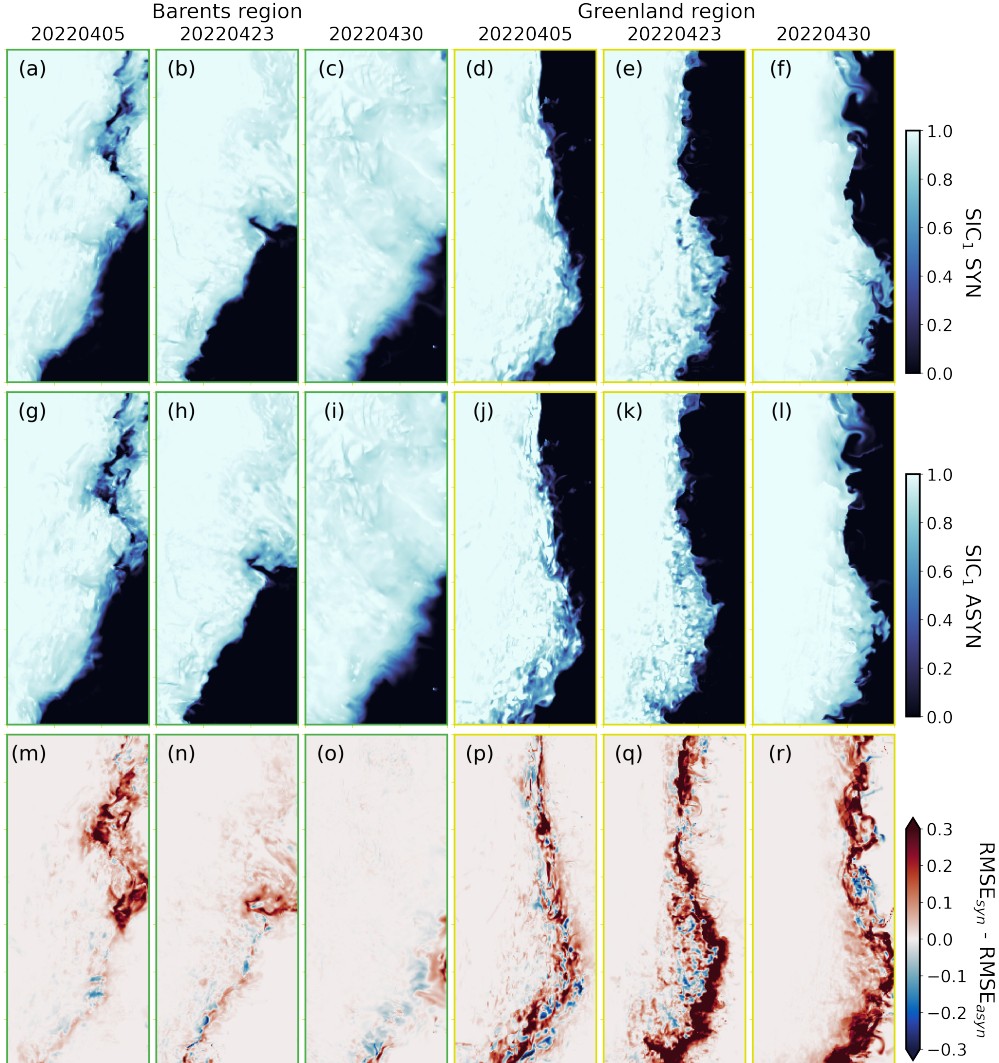

**Figure 11.** SIC fields of ensemble member 1 (a-f) from SYN and (g-l) ASYN experiments, and (m-r) RMSE differences ($RMSE_{syn}$-$RMSE_{asyn}$) for the 5, 23 and 30 April 2022. Model data correspond to model ensemble after assimilation for the first two dates, and to model forecast ensemble for the last date, being always member 1 represented in the SIC maps. RMSEs are computed applying Eq. (3), using the corresponding ensemble data and SIRANO L3 observations as inputs. Barents and Greenland maps are indicated by green and yellow colored axis respectively.

Figure 11 presents maps of SIC for ensemble member 1 for both DA experiments and RMSE difference maps ($RMSE_{syn}$-$RMSE_{asyn}$) from the model ensemble data within the two regions. For the first two dates (5 and 23 April 2022), the model data





used corresponds to the ensemble data after the EnKF analysis. For the last date (30 April 2022), the model data used is the

forecast ensemble data. Fields of SIC after analysis show lower SIC for both DA experiments compared to CTRL (see CTRL

fields in Fig. 7). The EnKF analysis removes SIC in the background ensemble aiming at creating open water areas present in

the observations (see SIRANO maps in Fig. 7). However, these SIC fields also present the history of the model run during the

inter-analysis periods. After the 7-day forecast period, on 30 April 2022, the fields of SIC present an increase of ice-covered

area in the two regions due to the freezing process generated by the model dynamics.

       The RMSE difference maps are obtained from the analysis ensemble data and SIRANO L3 observations using Eq. (3). These

maps show a higher RMSE for SYN (red areas) compared to ASYN in both regions. In the Barents region this is visible mainly

in the upper right zone where ASYN is able to deliver a larger correction with positive values ($>0.1$) indicated in the RMSE

difference maps. However, these differences do not increase throughout the assimilation period and fade away by the end of

the forecast period with an average difference close to zero. The improved performance by ASYN is more remarkable in the

Greenland region where strong positive differences ($>0.3$) are present along the ice edge and some pronounced differences

($>0.1$) in areas located in the ice covered areas. In this region, the RMSE differences increase through the assimilation period

with an average difference of $\sim0.02$ on the 3 April and of $\sim0.05$ on the 23 April. Furthermore, this positive RMSE difference

remains large throughout the 7-day forecast period, with an average value of $\sim0.04$ on the 30 April, still mainly around the ice

edge, which has shifted eastwards due to the freezing process in the model.

### 4.3  Ice-charts validation

Ice-chart data are used to perform an independent validation of the DA results. We first compute ice-classes maps from the

model (CTRL, SYN, and ASYN experiments) to compare them to ice-charts maps (see example in Fig. 5, Sect. 3.3). First,

we compare the experiments ice-classes maps to ice-charts on days where these are available in the period of 3-30 April

2022. In these comparisons, we include analysis states after the EnKF analysis and model states in the inter-analysis period.

Pearson correlation coefficients are computed from model ice-classes maps and available ice-charts for both regions (Barents

= B, Greenland = G). There is a better correlation with ASYN ($corr_B = 0.78$, $corr_G = 0.61$, $p < 1$) and SYN ($corr_B = 0.76$,

$corr_G = 0.57$, $p < 1$) data compared to CTRL ($corr_B = 0.72$, $corr_G = 0.47$, $p < 1$) data, with highest correlation values for

the ASYN experiment. Figure 12 shows modelled (x-axis) versus observed (y-axis) spatially averaged SIC computed from

model ice-classes and ice-charts maps. Because of the over-estimation of SIC in the model, the data cloud is always below the

perfect correlation line (gray line). SYN and ASYN experiments provide an improvement compared to the CTRL, with ASYN

yielding best results.

Then, we evaluate the 7-day forecast period separately by computing MADs between model ice-classes and ice-chart maps

for both regions. The obtained MADs are significantly lower ($\sim29\%$ lower for the entire domain) than the ones obtained

from the comparison with SIRANO ice-classes maps. This is due to ice-charts presenting higher SICs than SIRANO ice-

classes maps, in particular during the forecast period (see Sect. 3.3). Towards the end of the forecast period, on 29 April 2022,

the ASYN experiment shows a 14.3% improvement regarding the MAD compared to the SYN experiment in the Greenland

region; whereas a weaker improvement (2.5%) is provided in the Barents region. All these validation diagnostics indicate that




the ASYN experiment provides better results, with a better performance of this experiment in the Greenland region. Overall,
this ice-chart MAD validation is in agreement with conclusions obtained in the SIRANO comparisons in Sect. 4.2.

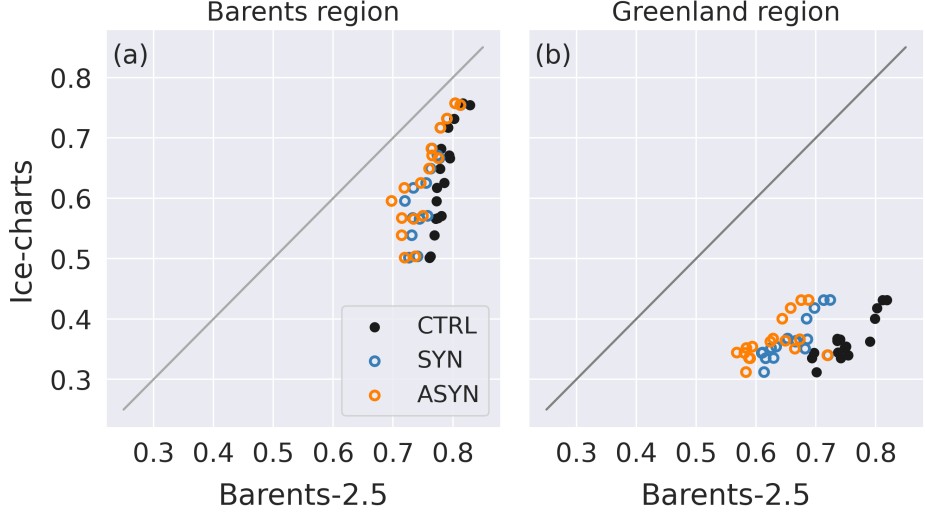

**Figure 12.** Spatial averages of ice-classes maps from *Barents-2.5* model (x-axis) and ice-charts (y-axis) for the (a) Barents and (b) Greenland
regions from CTRL (black), SYN (blue) and ASYN (orange) data.

# 5 Discussion

## 5.1 Asynchronous assimilation: why do we observe an improvement?

As demonstrated in Sec.4, the ASYN experiment outperforms the SYN experiment (see e.g. Fig. 10). The only difference be-
tween the experiments is the observations that are assimilated (L3 or L3U) and how they are assimilated (synchronously or
asynchronously).

To better understand the differences between the experiments SIC power spectra are computed to evaluate how the assimilation
modifies the variance at different spatial scales in the model fields. The spectra are calculated for the subregions indicated by
the dashed rectangles in Fig. 1 (see also Sect.3.3). The model spectra are computed from the background and analysis ensem-
ble on dates where EnKF analysis is performed. We also compute the spectra from the CTRL ensemble at the same dates as
a reference. The spectrum is first calculated for each ensemble member, then these spectra are averaged. The resulting spec-
tra are shown in Fig. 13a and Fig. 13b. Both regions show higher power in ASYN than in SYN at all spatial scales. Higher
power indicates that we have more detailed SIC fields with more spatial features at the respective wavelength. This is visible in
Fig. 11, where the SIC fields from ASYN present sharper spatial structures than the corresponding SYN fields. In the ASYN
analysis, we assimilate L3U observations which have a significantly higher power spectrum than L3 with ratios above 1.2 (see
Fig. 4) at scales below ∼55 km and ∼30 km wavelength for the Barents and Greenland regions respectively. Assimilating these
more detailed observations produces an updated model ensemble of SIC with higher spatial power at all scales. Background



power spectra have less spatial power than the analysis spectra but still remains significantly higher than the CTRL spectra. This indicates that during the inter-analysis period part of the power injected by the EnKF analysis is lost. This power reduction is due to the model re-freezing process which decreases the spatial variance in the SIC fields.

Ratios of ASYN to SYN spectra are shown in Fig. 13c and Fig. 13d. These ratios are always above 1, showing that the ASYN experiment presents more spatial power than the SYN, with larger ratios in the Greenland (ratio$_{max}$=3.8) than in the Barents (ratio$_{max}$=2.4) region. In general, the background ratio (dashed line) is lower than the analysis ratio (solid line) which means that ASYN suffers a stronger power reduction during the inter-analysis periods, due to an often stronger model freezing rate in the ASYN than SYN during these periods. In both regions, below ∼20-30 km wavelength, we observe a significant differ-

ence between the analysis and the background ratios. This indicates that for small wavelengths, the ASYN analysis produces a stronger increase of spatial power in the updated SIC states compared to SYN. In the Barents region there is a significant difference between background and analysis ratio above ∼100 km too, indicating that ASYN is injecting more power at large scales. This is related to the strong temporal variability of the SIC increments in this region, which affects all scales. In contrast, there is a steady temporal variability of the SIC increments in the Greenland region, and the ratio difference is located mainly

at small wavelengths (<30 km) where corrections by the EnKF are required.

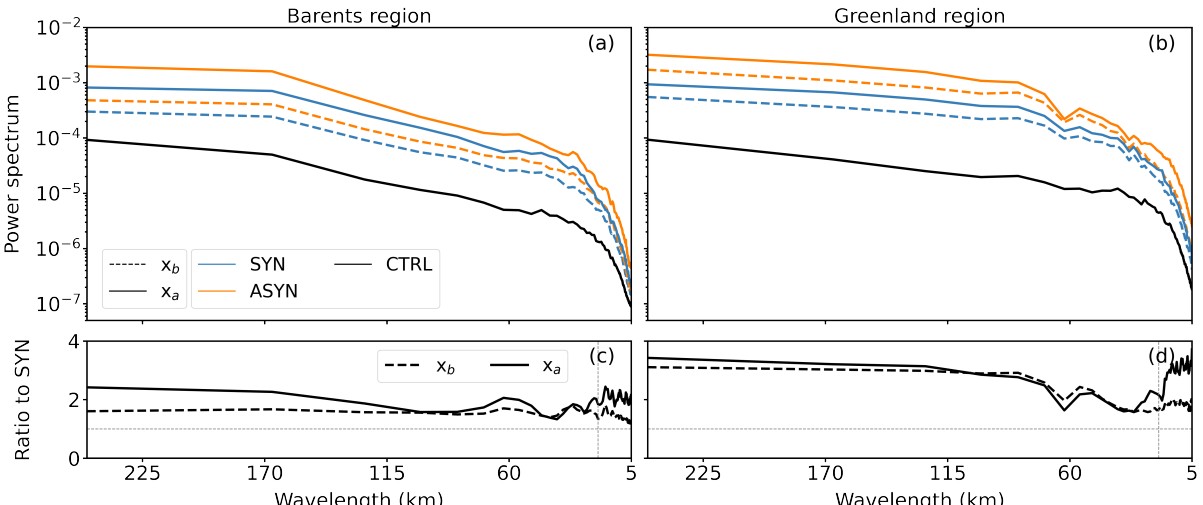

**Figure 13.** Power spectra computed from the CTRL (black lines), SYN (blue lines) and ASYN (orange lines) experiments from analysis (solid lines) and background (dashed lines) ensemble states in the (a) Barents and (b) Greenland regions; and (c, d) the respective ratios of ASYN to SYN spectra for each region.

The power spectra computed from the DA and CTRL experiments during the 7-day forecast period are shown in Fig. 14a and Fig. 14b. Both DA spectra have significantly higher power than the CTRL spectra with average ratios to CTRL of ∼1.4 for the Barents region and ∼7.4 for the Greenland region. These high ratios show that part of the power gained after the assimilation

period is still maintained during the forecast period, being more pronounced in the Greenland region. In the Barents region,



the ratio of ASYN to SYN spectra (see Fig. 14c) shows a value of ∼1 for all scales, indicating that the higher power provided by the ASYN experiment is lost after the forecast period. In contrast, the Greenland region presents large spectra differences between the two DA experiments. High ratios (>3) of ASYN to SYN spectra are present for scales larger than 100 km, with a slight ratio increase (∼1.5-2) for wavelengths below 30 km. This indicates that ASYN keeps the higher spatial power compared

to SYN for both large and small scales during the forecast period in the Greenland region. We expect this ratio to progressively decrease for longer forecast periods and become similar to the Barents ratio. Since the Greenland region has a slower model re-freezing process, it takes more days to significantly decrease the spatial power in the SIC forecast fields and thus, to converge towards the CTRL spectrum.

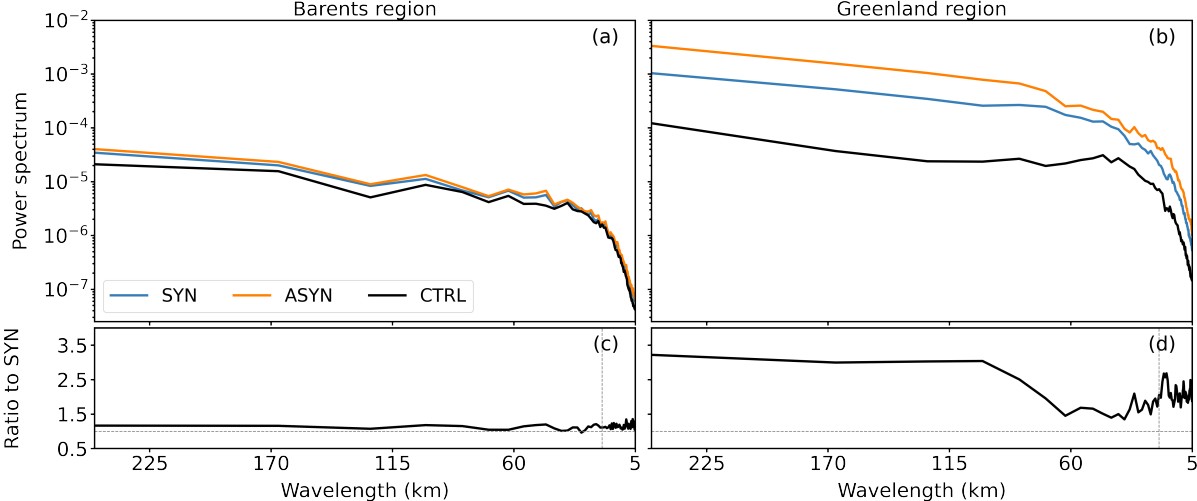

**Figure 14.** Power spectra computed from the CTRL (black lines), SYN (blue lines) and ASYN (orange lines) experiments from forecast states (7-day forecast period) in the (a) Barents and (b) Greenland region; and (c, d) the respective ratios of ASYN to SYN spectra for each region.

Figure 15 presents ratios of ASYN to SYN of the (i) number of assimilated observations; (ii) the DFS observation metric; and

(iii) the DASS for each EnKF analysis in the entire region. The ratio of the number of assimilated observations shows values of ∼3 during the entire assimilation period indicating that in average 3 times more observations are assimilated in the ASYN analysis. The higher quantity of observations translates into a stronger assimilation by the ASYN analysis as indicated by a DFS ratio above 1. On average, this DFS ratio is 1.8, meaning that indeed the assimilation is stronger but below the observation ratio. This is due to the setup of the observation errors and the reduced spread after assimilation. The stronger assimilation by

the ASYN analysis provides a more accurate system (1.3 times more accurate on average) as indicated by the DASS ratio. A DASS ratio above 1 means that the information assimilated is stronger than the double penalty-effect, which usually penalises the forecasts with higher effective resolution.



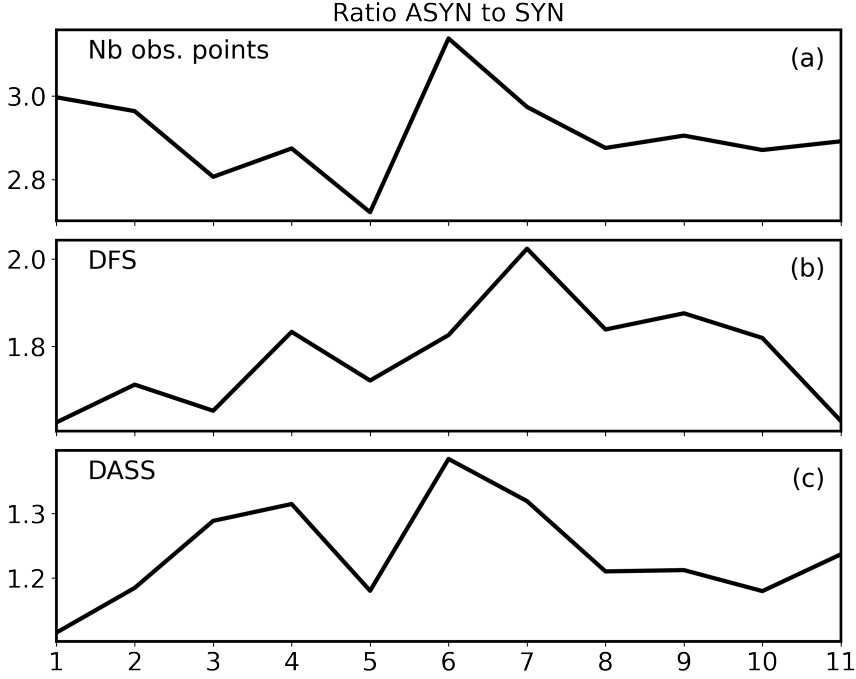

**Figure 15.** ASYN to SYN ratios of (a) number of observation points used in the assimilation; (b) DFS (see Sect. 2.3); and (c) DASS (see Fig. 10) computed in the entire region.

The improved results obtained by ASYN compared to SYN are therefore due to: (i) a higher presence of spatial power in the L3U swaths; (ii) a higher number of observation points used in the ASYN assimilation leading to larger DASS. In the presented DA experiments, the localisation radius is set to 20 km. This seems like a fair choice, as this is the scale where the L3U power approaches a level of twice the L3 power (ratio≥1.5, see Fig. 4). Thus, an EnKF analysis using a 20 km localisation radius takes this power difference into account. A larger localisation radius is not expected to include more information regarding this higher power contained in the L3U observations, and it could be detrimental to the analysis with regard to the DFS metric as we have a small ensemble. A much lower localisation radius (for instance 10 km) can on the other hand lead to valuable information on spatial scales larger than the localisation radius present in the observation not being optimally used during the EnKF analysis step, as the observation points are unable to represent spatial scales larger than the localisation radius.

## 5.2 Regional dependence of the assimilation results

Although the ASYN analysis provides a better correction than the SYN analysis in both regions, the inter-analysis period influences the experiments differently within each region. The inter-analysis period deteriorates the experiment performance in the Barents region, whereas this is not the case in the Greenland region. Thus, the model itself strongly affects the regional dependence of the experiment outcomes.

Figure 16 presents fields of SIC innovation $[y(t) - Hx_b(t)]$, increment $[x_a(t) - x_b(t)]$ and background spread from the ASYN





analysis. Fields are shown for two different dates (5 and 23 April 2022) and for both regions. The innovations show a large difference in the Barents region, which is related to the high temporal variability in the observations as indicated in Fig. 7. The ensemble spread in the Barents region is significantly lower ($\sim$2 times lower) than in the Greenland region, with low spread areas often corresponding with high innovations areas. The EnKF is unable to produce large SIC increments ($>$0.1) in these areas. Due to the continuous spread reduction by the EnKF analysis, this issue becomes more prominent throughout the assimilation period. This can be observed in the lower spread and increment values on the 23 compared to the 5 April. The Greenland region, however, presents innovation fields with similar spatial distribution on the 5 and 23 April, while the innovation values are decreased during this period. SIRANO observations present low temporal variability in this region compared to the Barents region (see Fig. 7), and innovations therefore have similar spatial distribution during the assimilation period. The Greenland region is characterized by stronger ensemble spread which is located around the ice edge in accordance with areas of high innovation. The EnKF can therefore always produce large increments ($>$0.1) in these areas. In conclusion, the Greenland region has the advantage of having large spread correctly located as well as low temporal variability of the innovations. Each EnKF analysis thus aims at getting closer to a solution similar to the previous analysis, and the error reduction of the experiments is accelerated accordingly.

The spread differences between SYN and ASYN are shown in Fig. 16m-p. In the Barents region, the differences are quite low for both dates with the highest differences localised in areas of high ASYN spread. Thus, SYN and ASYN have similar values and spatial distribution of spread in this region. In contrast, the Greenland region shows significantly larger spread differences on the 23 than 5 April. Furthermore, the spread differences in this region show a clear spatial pattern: the SYN spread presents higher values in the eastern area of the region, whereas the ASYN presents higher values in the western area of the region. Due to stronger ice removal in ASYN (see Fig.6), a larger open water area is created with a strong westward displacement of the ice edge. As high spread is located around the ice edge, there is a corresponding westward shift of high spread in ASYN compared to SYN. As this westward shift is in agreement with the observed location of the ice edge, this spread difference relates to ASYN being able to perform this correction at a faster rate than SYN.

Due to the strong innovation variability together with a lack of high spread in large innovation areas, the differences between the two DA experiments do not accumulate with time in the Barents region, whereas that is the case for the Greenland region. The variability of innovations is therefore a crucial aspect to consider, as it can strongly hamper the experiment results.


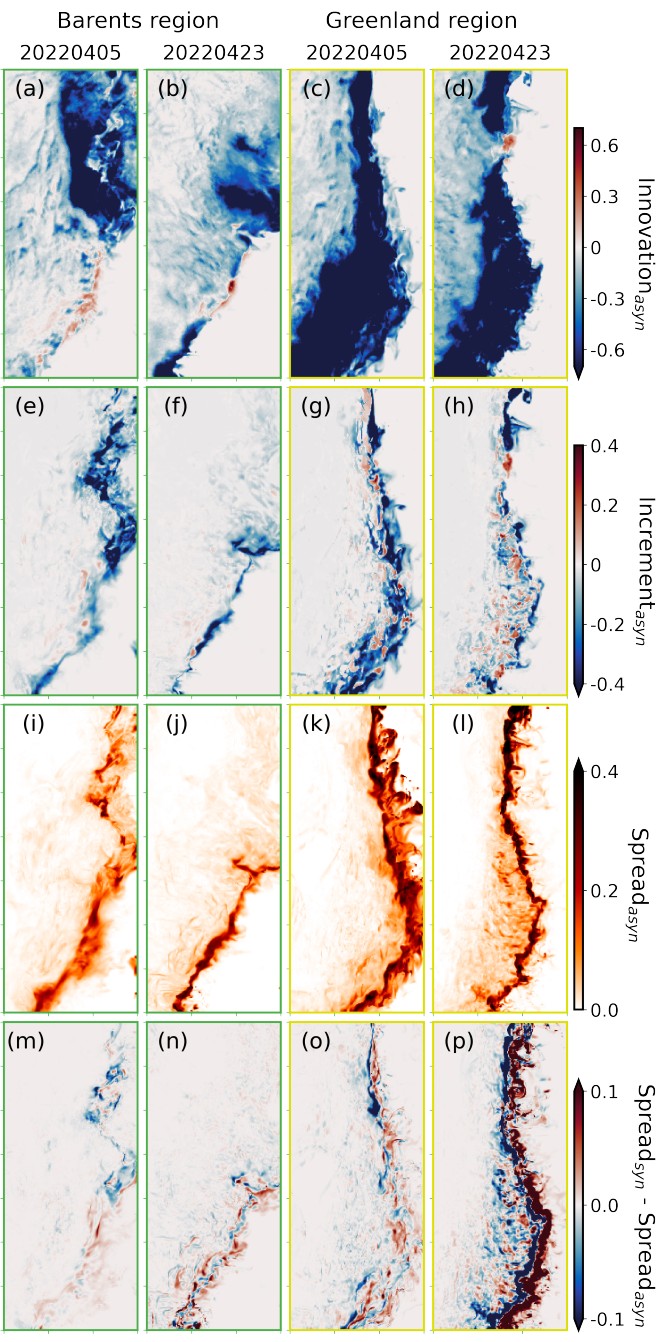

**Figure 16.** Maps of SIC (a-d) innovation $[y(t) - Hx_b(t)]$; (e-h) increment $[x_a(t)-x_b(t)]$, and (i-l) spread from the background ensemble, with respect to the ASYN experiment; and maps of (m-p) the difference of background model spread between SYN and ASYN. Maps shown correspond to model data at days 5 and 23 April 2022 in the Barents and Greenland regions. Barents and Greenland maps are indicated by green and yellow colored axis respectively.



Figure 17 presents the total change of SIC (%) after each EnKF analysis ($[x_a(t) - x_b(t)]/x_b(t)$) and after each following inter-analysis period ($[x_b(t+48\,hours) - x_a(t)]/x_a(t)$). A negative change indicates a SIC reduction, and a positive change a SIC increase. Every EnKF analysis produces negative SIC changes (see Fig. 17a). Throughout the assimilation period, this SIC reduction weakens due to the spread reduction induced by the EnKF (see time series of spread in Fig. 6c). The ASYN analysis presents stronger SIC reduction than SYN, with larger reductions in the Greenland region. This is in accordance with results presented in Sect. 4. After the inter-analysis period (see Fig. 17b), we generally observe positive SIC changes due to the model generation of sea ice with only a few negative changes. The four first inter-analysis periods present quite different performance of the DA experiments within each region. The Greenland region presents weaker positive changes ($\sim 2\%$) with a negative change on the fourth period. This means that the model sea ice generation is quite low with a model SIC reduction on the fourth period. In contrast, the Barents region presents larger positive changes ($\sim 4\%$) during these four periods indicating that the model is always re-freezing areas where SIC was decreased by the EnKF analyses. The ASYN changes are larger than the changes in SYN which translates into larger SIC increases (or lower SIC decreases) for the ASYN experiment. In the Barents region, the difference between SYN and ASYN changes is larger. This indicates a stronger model re-freezing response in the ASYN experiment which entails a loss of the improvement after the EnKF analysis. This difference between SYN and ASYN changes is weaker in the Greenland region, and thus, the improvement gained by the ASYN analysis is better maintained after the inter-analysis period compared to the Barents region. As can be observed in Fig. 6, the CTRL experiment presents quite constant mean SIC ($\sim 0.8$) in the Barents region, whereas in the Greenland region there are stronger fluctuations. With respect to the first four assimilation cycles, the CTRL experiment presents a SIC decrease (from 0.78 to 0.72) in the Greenland region. This model behaviour helps maintain the SIC changes induced by the assimilation in this region.

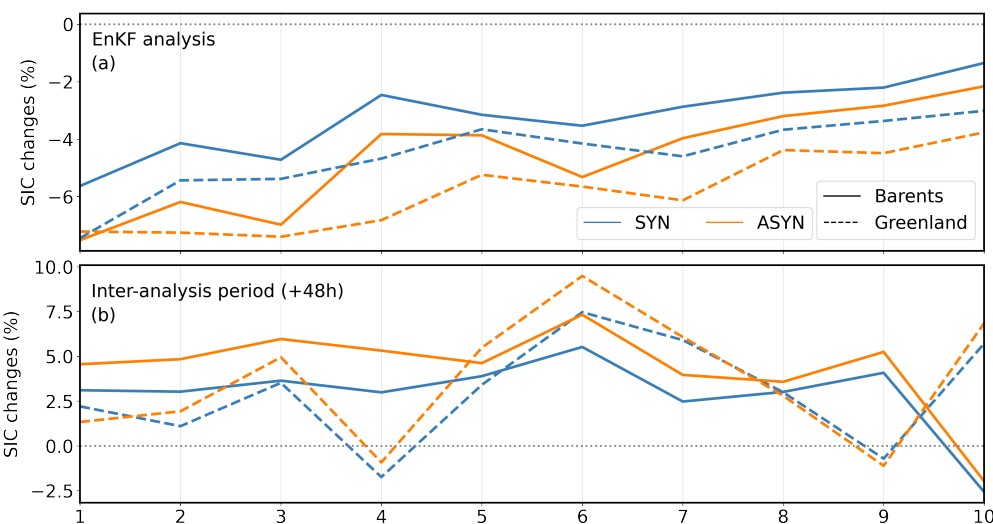

**Figure 17.** SIC changes (%) (a) after each EnKF analysis, and (b) after each inter-analysis period (+48 hours) in the Barents (solid lines) and Greenland (dashed lines) regions for both SYN (blue) and ASYN (orange) experiments. Positive changes indicate an increase of SIC, and negative changes a decrease of SIC.



# 6 Conclusions

In this study we investigate the benefits of assimilating individual satellite swaths instead of daily means of SIRANO SIC
in the *Barents-2.5* numerical model using the EnKF. The comparison between the assimilation of individual swaths (ASYN
experiment) and daily means (SYN experiment) of satellite SIC shows better performance by ASYN: at the end of the assim-
ilation period we have a domain-wide MAD improvement of 10% in ASYN compared to SYN, and of 7% at the end of the
forecast period. The DASS shows that the EnKF analysis provides better results when assimilating the swaths rather than the
daily means. An independent validation using ice-chart data corroborates the ASYN improvement.

Two smaller regions are selected to evaluate with more detail the performance of the assimilation experiments in the ice
edge area. The EnKF analysis performs well in both regions, with positive values of DASS by both DA experiments and with a
better performance by the ASYN analysis in both regions. However, in the Barents region this improvement is rapidly reduced
after 48 hours of model run, and vanishes after a 7-day forecast period. The Greenland region, on the other hand, shows a
pronounced improvement provided by the ASYN assimilation which is well maintained throughout the forecast period. These
results show that there is a regional dependence of model skill in our experiments due to different sea ice conditions present in
the observations and model fields in the two regions.

The SIC reduction by the DA experiments in both regions entails a SIT reduction as well as increases of SST and salinity
which are both stronger in ASYN than SYN. As more ice is removed in ASYN, the production of new ice after analysis
provides a lower SIT than in SYN. The SST and salinity increases are a response of the model dynamics to the new open water
areas where different mixing processes take place, leading to a stronger mixing in ASYN. Therefore, in addition to an improved
SIC forecast, the assimilation of individual swaths can potentially have a positive impact in the forecast of other variables such
as SIT, SST and salinity.

Although we observe a regional dependency of the results, the DA experiments clearly indicate that the assimilation of
the individual swaths provides a better forecast than the assimilation of the daily means, with an average positive impact
even after a 7-day forecast. When assimilating individual swaths, more observations points are being assimilated in the EnKF
analysis. This increases the DFS observation metric indicating that there is a stronger impact of the observations within the
same localisation radius, which leads to larger DASS values for the ASYN experiment. We note that reducing the observations
errors of the daily means could have had the same effect, however these daily means would still have smoother features than
the individual swaths. Also, reducing observation errors might entail a too strong spread reduction by the EnKF which, after a
few cycles, can negatively impact the experiment results.

The power spectra computed from the swaths and daily means show that the swaths contain stronger spatial power, meaning
that there are more spatial features and details in the SIC fields, in particular for structures below 50 km of wavelength. The
power spectra from the DA experiments show that the spatial power increases after assimilation and always stays higher than
the CTRL power even after the inter-analysis period. This power increase occurs at a higher rate in the ASYN assimilation
in both the Barents and Greenland region. The power spectra computed during the forecast period shows a loss of the power
gained in the Barents region compared to the CTRL. The spectra in the Greenland region, on the other hand, maintain most of



the power during the forecast period. This is again related to the model dynamics, with a faster freezing in the Barents region during the period of the study. The variance of the homogeneous ice-coverage created by this freezing process is quite low, entailing a spatial power loss. The model dynamics in the Greenland region is beneficial (mainly during the first half of the assimilation period) for maintaining the reduced sea ice cover, and creates a strong difference between the DA experiments and the CTRL as well as between the DA experiments themselves. These differences are maintained throughout the forecast period and hence are transferred into the power spectra, with the variance injection by the EnKF analysis still persisting after a 7-day forecast. The abundance of small scale features in the ASYN experiment could have penalized the skills score because of the statistical double-penalty effect, but this was not the case here, indicating real practical skills in the assimilation of swaths.

In order to have an improved performance of the ASYN experiment in all regions, the model spread should be in accordance with the observations variability. In the period covered by our experiments, the sea ice model tends in general to freeze up too much and creates homogeneous ice-covered areas. This model bias creates negative impact in the Barents region where observations show strong spatial changes within short time spans. In order to have sufficient model uncertainty, the ensemble members need to spread further from one another than what is currently the case. To achieve better spread, the use of different CICE numerical schemes and parameters in different members could be explored following investigations by Massonnet et al. (2014) and Urrego-Blanco et al. (2016). Another alternative would be to explore the atmospheric forcing and better understand the air-ice interactions performed by the sea ice model in order to evaluate what atmospheric variables could be tuned to increase the model spread.

In conclusion, this work demonstrates that the assimilation of individual swaths of SIC provides more accurate modelled SIC forecasts. More observations are assimilated in the swaths assimilation, which are used to update the model at the correct time intervals, leading to a more accurate assimilation system with model forecasts presenting lower deviations to the observations. On top of that, the satellite swaths contain more spatial details, in contrast to the daily means with a blurred spatial resolution due to the temporal averaging of the swaths. Thus, the assimilation of swaths provides corrections at small spatial scales and thereby increases the spatial power in the model forecasts compared to the case of assimilation of daily means. This increase of spatial power indicates a higher spatial variability in the analysis states compared with the background states, which is due to the assimilation aiming at reducing the background SIC characterized here by highly homogeneous ice-covered areas compared to the observations. One major advantage of the swaths assimilation in operational systems is that there is no need to wait for the daily means availability in order to update the model forecast. This point is particularly important in the context of coupled atmosphere-ocean-sea ice forecast systems, as numerical weather prediction commonly applies shorter update cycles than ocean forecast systems. Furthermore, the computational cost of the ASYN EnKF analysis is not significantly higher than the SYN analysis: in our experiments the ASYN analysis takes around 30 seconds more than SYN, with only a higher number of input files needed leading to a slightly larger storage demand (on average 150 M more per analysis).

*Code and data availability.* A source code for the coupled ROMS/CICE ocean and sea ice model is available through https://doi.org/10.5281/zenodo.5067164 tagged as version 0.4.1 of the METROMS repository. The source code for the EnKF-C v.2.9.9 is obtained from





https://github.com/sakov/EnKF-C.git, commit *7eea4d8* as of Jul 8, 2021. Ice-chart data are retrieved from CMEMS https://doi.org/10.48670/
       moi-00128 and are available as graphic maps and through an API at https://cryo.met.no/en/latest-ice-charts. Model data from CTRL, SYN
       and ASYN experiments as well as SIRANO data will be made available through the MET Norway thredds server. Specific configuration and
       grid files for *Barents-2.5* as well as EnKF-C parameter files used for these experiments will be archived at zenodo.



## Appendix A



**Figure A1.** SIRANO individual satellite swaths (L3U) on the 2 April 2022.



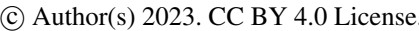

*Author contributions.* MDM: conceptualization, methodology, investigation and software (experiments setup and implementation), formal data analysis and visualization, writing – original draft preparation and writing - review & editing. AKS: conceptualization, methodology, formal data analysis, supervision (Work Package leader), writing - review & editing. TL: conceptualization, methodology, funding acquisition, project administration (project manager), supervision, software (SIRANO observations), writing - review & editing. LB: formal data analysis, writing - review & editing. YG: software (ice-classes maps), writing - review & editing. SCI: software (power spectrum), writing - review & editing. JR: software (SIRANO observations), writing - review & editing.

*Competing interests.* There are no competing interests associated with this work.

*Acknowledgements.* We acknowledge funding by the Norwegian Research Council of Norway, grant No. 302917 (SIRANO). This study has been conducted using E.U. Copernicus Marine Service Information, i.e. model data as boundary and initial conditions[10] and ice-charts data[11] in the validation. We acknowledge the R&D contribution of ESA CCI Sea Ice and EUMETSAT OSI SAF projects to the preparation of the regional SIC product. Regarding the production of the SIRANO observations, we acknowledge the technical support provided by Atle Macdonald Sørensen. A particular thanks to Pavel Sakov for the technical support and help on the use of the EnKF-C software. We want to thank Sindre Fritzner for all the initial guidance on how to set up the EnKF-C with the *Barents-2.5* model configuration. We acknowledge the contributions of Jiping Xie and Yue Ying from NERSC, through all the interesting and helpful discussions about the EnKF and the assimilation results.

---

[10]https://doi.org/10.48670/moi-00001
[11]https://doi.org/10.48670/moi-00128





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
