# Peer review of "Assimilation of satellite swaths versus daily means of sea ice concentration in a regional coupled ocean-sea ice model"

_The Cryosphere, 2023_

## Author Comment (AC1)

**Responses to comments from referee #1**

**Summary:** In this study, a regional configuration of the Barents Sea modeling system composed of ROMS and CICE 5.1.2 is used to study the impact of the assimilation of swath AMSR2 sea ice concentration data versus daily means of SIRANO sea ice concentration data. Particular focus was given to sub-regions with the 2.5km domain for the Barents and Greenland Sea. Two sets of atmospheric forcing are used to introduce ensemble spread: 1) Integrated Forecast System developed at ECMWF which provided members 1-5, and MET-AA which provided member 6. The EnKF is used as the data assimilation system in this study. Three experiments are performed: 1) Control run without DA, 2) synchronous assimilation using SIRANO data, and 3) asynchronous assimilation using AMSR2 swath data. This study found that the assimilation of the swath AMSR2 sea ice concentration led to a 10% improvement in the MAD at the end of the assimilation period and 7% improvement at the end of the 7-day forecast period.

This is a very thorough and well written paper. I only have minor comments listed below. I recommend publication.

*Response: We thank the reviewer for the positive feedback and comments on the manuscript. Following the comments and suggestions, we provide answers and point out the modifications performed in the manuscript below.*

**General Comments:**

Line 304: Rephrase "As CICE does not…" to something like "While CICE 5.1.2 used in this study does not differentiate between stationary ice attached to land, CICE6 includes a landfast ice parameterization (https://zenodo.org/record/7419531).

*Response: We have clarified in the text that this applies to the CICE version used in this study and not to all CICE versions. Text added in Line 304, "As the CICE version used in this study does not differentiate between stationary ice attached to land and ice that is floating freely, the FI of the ice-charts maps is considered as VCDI in the validation."*

Fig. 5: Why was this particular month chosen (April 2022), with the Easter Holiday occurring mid-April? You lose data for 5 days (April 14-18) versus the typical 2 days on weekends? Since you should have the SIRANO data, I suggest you add that information for the Barents, Greenland, and Entire region to the plot.

*Response: Yes, we agree that having the Easter holiday within the period of study was not beneficial for the ice-chart comparison. Initially, the period of study was chosen to be on the same time as an oceanographic cruise that took place in April 2022 in case we wanted to compare with the collected in-situ measurements. However, the measurements were too sparse in order to have a meaningful comparison with our experiments. As the most critical period for the ice-chart validation is the 7-day forecast, during which there were no holidays, we decided to keep April as the study period. Figure 5d's objective is to show the ice-classes maps from*

*SIRANO and ice-charts used in the validation presented in Section 4.3. SIRANO ice-classes map data was not added in the figure for the periods where ice-charts are not available to avoid confusion on what data was used in the validation. As SIRANO time series are already shown in Fig.6a-b, we rather keep Fig. 5 as it is.*

Figure 9 caption: "Mean Absolute Difference" is defined as Mean Absolute Deviation on line 166. Please make correction.

*Response: changed to "Mean Absolute Deviation".*

Lines 443-450: Do you have any graphics or tables to support your (29% lower), (14.3% improvement) statements?

[Figure]

*Fig R1. Time series of MAD computed between ice-charts (solid) and SIRANO (dash-dotted) and model ice-classes maps in the (a) entire, (b) Barents and (c) Greenland regions during the forecast period.*

*Response: The MADs are computed from SIRANO and ice-chart ice-classes maps and model data. The time series of MADs during the 7-day forecast period is shown in Figure R1 for the entire region, and Barents and Greenland subregions. On average, the ice-chart time series for*

*the entire region presents a 29% lower MAD than the SIRANO time-series. Regarding the ice-charts time series, ASYN shows an improvement of 12.2% compared to the SYN experiment at the end of the 7-day forecast period in the Greenland region ($MAD_{SYN}$=0.2933, $MAD_{ASYN}$=0.2575). In contrast, the ASYN improvement is much weaker (1.5%) at the end of the forecast period in the Barents region ($MAD_{SYN}$=0.0929, $MAD_{ASYN}$=0.0915). These two numbers have been updated in the manuscript as previous numbers (14.3%, 2.1%) corresponded to the end of the assimilation period instead of the forecast period. Figure R1 has been added to the Appendix of the manuscript in order to clarify the origin of these computed values.*

---

## Author Comment (AC2)

**Responses to comments from referee #2**

Review on "Assimilation of satellite swaths versus daily means of sea ice concentration in a regional coupled ocean-sea ice model", by Marina Durán Moro, Ann Kristin Sperrevik, Thomas Lavergne, Laurent Bertino, Yvonne Gusdal, Silje Christine Iversen, and Jozef Rusin, submitted for publication in The Cryosphere.

**General comments :**

The paper presents experiments where sea ice concentration observations (retrievals) are assimilated in a coupled ice-ocean model. The assimilation follows the method of the Ensemble Kalman Filter (EnKF). The goal of the study is to measure the impact of assimilating individual satellite observation swaths as opposed to assimilating daily average sea ice concentration derived from the same data set. The paper is very well written.

Three runs are performed. The first is a control run without data assimilation (or free run), the second run assimilates daily average sea ice concentration every 2 days (SYN), and the third run assimilates the sea ice concentration retrievals from satellite observation swath (ASYN).

*Response: We appreciate the comments and suggestions the reviewer provides on this manuscript. We address each of these comments and present answers and a description of the changes realized in the manuscript below.*

One concern I have is the use of the mean error of the ensemble members instead of the error of the ensemble mean. I am wondering about this choice, since the ensemble spread should correspond to the error of the ensemble mean, and not to the mean error of the ensemble members. Please comment.

*Response: We agree that the ensemble spread should estimate the error of the mean instead of the mean error of the ensemble members. We therefore have updated the RMSE equation in Section 2.4 defined as RMSE^2 = (<X>-Y)^2 with <X> the ensemble mean and Y the observation. The computed RMSE values have been updated accordingly in the text as well as figures 10, 11 and 15. The associated reduction of the RMSE improves the agreement between the ensemble dispersion and the actual errors and puts our results in a more favorable light with a consistent lower (~4%) RMSE compared to the non-updated RMSE values.*

**Specific comments:**

Line 144: Please provide a reference for DFS, maybe Cardinali et al. (2004)

*Response: The citation of Cardinali et al (2004) has been added in Line 144.*

Line 223: What is the "K-factor" ? Is the observation-error covariance matrix diagonal ? If the observation-error variances are increased by a factor 20 (R-factor = 20), what is the based sea ice concentration observation-error variance ? Is the observation-error variance homogeneous ?

Some of these questions are answered later in the text in detail, but it could be nice to have general info upfront for the readers.

*Response:*

*The K-factor is an adaptive moderation factor which reduces the impact of observations incompatible with the background/priori. This is performed by gradually increasing the observation error as a function of the magnitude of the innovation.*

*The observation-error covariance matrix (R) is assumed to be diagonal by the EnKF-C. SIRANO observation errors present weak spatial correlations ("systematic errors" discussion in Lavergne et al, 2019), and we believe that the assumption of a diagonal R matrix is applicable.*

*The R-factor corresponds to a scaling coefficient which multiplies the observation error variances, leading to an increase of the observation impact when R-factor decreases. The R-factor multiplies the variances in the observation-error matrix R in equation (6) defined in Sakov et al. (2010) leading to scaled ensemble observation anomalies.*

*The SIRANO observation errors are high in the ice-edge area, decreasing with the distance to it as shown by Fig.3d-f in the manuscript. The observation-error variance is therefore not spatially homogeneous.*

*We have added a description of the K-factor in Line 141. In lines 226 to 228, we have extended the discussion regarding the EnKF parameters. More technical details on these parameters are presented in the EnKF-C documentation (Sakov, 2014).*

Line 333: The fact that the ensemble spread does not reduce to zero does not mean that the system is well tuned. It is a necessary but not a sufficient condition. For a well-tuned system, we would expect some overlap of the shadows of the model spread and the observations uncertainty in figure 6.

*Response: We agree that the ensemble is not well tuned and rather use the expression "not collapsing". We suggest further improvements of the ensemble in the conclusions (Section 6), including an objective measure of the ensemble spread with the RCRV (Reduced Centered Random Gaussian) introduced by Candille et al (2015).*

**Technical corrections**

Line 279: "are are" typo, repetition

*Response: corrected in manuscript.*

Line 367: "with a slight larger" should be "with a slightly larger"

*Response: corrected in manuscript.*

Line 370: SSS should be replaced with salinity since that is the term used throughout.

*Response: corrected in manuscript. SSS is replaced by sea surface salinity (SSS).*

Figure 8: Replace SSS with salinity or define SSS = Sea Surface Salinity

*Response: SSS is kept in Figure 8, previously in the text (Line 370) we write "sea surface salinity (SSS)".*

**References:**

*Candille, G., Brankart, J.-M., and Brasseur, P.: Assessment of an ensemble system that assimilates Jason-1/Envisat altimeter data in a probabilistic model of the North Atlantic ocean circulation, Ocean Science, 11, 425–438, https://doi.org/10.5194/os-11-425-2015, 2015.*

*Cardinali, C., Pezzulli, S., and Andersson, E.: Influence-matrix diagnostic of a data assimilation system, Quarterly Journal of the Royal Meteorological Society, 130, 2767–2786, https://doi.org/10.1256/qj.03.205, 2004*

*Lavergne, T., Sørensen, A. M., Kern, S., Tonboe, R., Notz, D., Aaboe, S., Bell, L., Dybkjær, G., Eastwood, S., Gabarro, C., Heygster, G., Killie, M. A., Brandt Kreiner, M., Lavelle, J., Saldo, R., Sandven, S., and Pedersen, L. T.: Version 2 of the EUMETSAT OSI SAF and ESA CCI sea-ice concentration climate data records, The Cryosphere, 13, 49–78, https://doi.org/10.5194/tc-13-49-2019, 2019*

*Sakov Pavel, Geir Evensen & Laurent Bertino (2010) Asynchronous data assimilation with the EnKF, Tellus A: Dynamic Meteorology and Oceanography, 62:1, 24-29, DOI: 10.1111/j.1600-0870.2009.00417.x*

*Sakov, P.: EnKF-C user guide, https://doi.org/10.48550/ARXIV.1410.1233, 2014*